# COMPOSITE OPTIMIZATION WITH ERROR FEEDBACK: THE DUAL AVERAGING APPROACH

**Yuan Gao**$^{*\dagger\ddagger}$    **Anton Rodomanov**$^{*\ddagger}$    **Jeremy Rack**$^{*\dagger\ddagger}$    **Sebastian U. Stich**$^{*\ddagger}$

## ABSTRACT

Communication efficiency is a central challenge in distributed machine learning training, and message compression is a widely used solution. However, standard Error Feedback (EF) methods (Seide et al., 2014), though effective for smooth unconstrained optimization with compression (Karimireddy et al., 2019), fail in the broader and practically important setting of composite optimization, which captures, e.g., objectives consisting of a smooth loss combined with a non-smooth regularizer or constraints. The theoretical foundation and behavior of EF in the context of the general composite setting remain largely unexplored. In this work, we consider composite optimization with EF. We point out that the basic EF mechanism and its analysis no longer stand when a composite part is involved. We argue that this is because of a fundamental limitation in the method and its analysis technique. We propose a novel method that combines *Dual Averaging* with EControl (Gao et al., 2024a), a state-of-the-art variant of the EF mechanism, and achieves for the first time a convergence analysis for convex composite optimization with error feedback that matches the best-known results in the uncomposite setting. Along with our new algorithm, we also provide a new and novel analysis template for inexact dual averaging method, which might be of independent interest. We also provide experimental results to complement our theoretical findings.

## 1 INTRODUCTION

Gradient methods, and in particular, distributed gradient methods, are the workhorse of modern Machine Learning. In this work, we consider a simple yet powerful extension of the basic optimization problem, namely, the composite optimization problem:

$$\min_{\mathbf{x} \in \mathrm{dom}\,\psi} \{F(\mathbf{x}) \coloneqq f(\mathbf{x}) + \psi(\mathbf{x})\},$$

where $f \colon \mathbb{R}^d \to \mathbb{R}$ is smooth and $\psi \colon \mathbb{R}^d \to \mathbb{R} \cup \{+\infty\}$ is a composite part. The composite optimization problem is ubiquitous in machine learning, and it covers a wide range of variants of the vanilla optimization problem, for example, regularized machine learning (Liu et al., 2015), signal processing (Combettes & Pesquet, 2010), and image processing (Luke, 2020). Since $\psi$ can take on the value of infinity, it also naturally covers the constrained optimization problem.

The sizes of datasets and models in modern Machine Learning have been growing rapidly, leading to unique challenges in the training process and the need for optimization algorithms tailored to these new settings. The distributed optimization paradigm has become a necessity due to the fact that one simply does not have the capacity to accumulate the entire dataset while training modern ML models. One of the most popular setups is to distribute the data across multiple clients/workers, and coordinate the model update on one server. Many of the recent breakthrough models are trained in such a setup (Shoeybi et al., 2019; Wang et al., 2020; Ramesh et al., 2021; 2022).

One of the main bottlenecks in scaling up distributed training is the *communication cost*. Transmitting the full large model updates between clients and the server can be prohibitively expensive when performed naively (Seide et al., 2014; Strom, 2015). One of the most popular practical remedy is *communication compression* with *contractive compression* (Definition 2.2) (Lin et al., 2018; Sun et al.,

---

$^{*}$CISPA Helmholtz Center for Information Security, Germany.

$^{\dagger}$Universität des Saarlandes, Germany.

$^{\ddagger}$\{yuan.gao, anton.rodomanov, michael.rack, stich\}@cispa.de

2019; Vogels et al., 2019). Contractive compressions are potentially biased, and naive aggregation of these biased compressed updates can lead to divergence (Beznosikov et al., 2020). In the classical setting when $\psi \equiv 0$, one of the most basic and popular families of methods that are used to rectify this issue in practice is the **Error Feedback (EF)** mechanism (Seide et al., 2014; Stich et al., 2018; Vogels et al., 2019; Ramesh et al., 2021). Due to its vast practical importance, EF mechanism has attracted significant interests in the theory community as well, where many works, though restricted to $\psi \equiv 0$, have attempted to theoretically explain the effectiveness of EF (Stich et al., 2018; Karimireddy et al., 2019; Stich & Karimireddy, 2020) or derive variants of EF that enjoy better theoretical properties than the original form (Fatkhullin et al., 2023; Gao et al., 2024a).

However, in the composite setting, the situation becomes much more complex, and the theory is much less developed. Existing works in the composite setting either impose some further restrictions on the objective (Islamov et al., 2025), cannot handle stochastic gradients (Condat et al., 2022), or have suboptimal rates (Qian et al., 2020).

The goal of our work is to address the general composite setting for the EF mechanism. We develop novel algorithmic and analytical tools, and we are the first to obtain rates for EF in the convex composite setting that match the uncomposite counterpart. We achieve the

$$\mathcal{O}\left(\frac{R_0^2 \sigma^2}{n\varepsilon^2} + \frac{R_0^2 \sqrt{\ell}\sigma}{\delta^2 \varepsilon^{3/2}} + \frac{\ell R_0^2}{\delta\varepsilon}\right),$$

convergence rate, matching the rates of state-of-the-art EF variants when $\psi \equiv 0$.

## 1.1 THE CLASSIC EF AND VIRTUAL ITERATION

Assuming that $\psi \equiv 0$, let us recall the classic EF mechanism and the main tool that is used to analyze it, the virtual iteration framework (Mania et al., 2017), to understand its drawbacks. On a high level, we consider an update rule of the form $\mathbf{x}_{t+1} = \mathbf{x}_t - \frac{1}{\gamma}\hat{\mathbf{g}}_t$, where $\hat{\mathbf{g}}_t$ is some estimate of the true gradient $\mathbf{g}_t = \nabla f(\mathbf{x}_t)$. EF provides a way to construct such an $\hat{\mathbf{g}}_t$ when the gradient information can only be communicated after being compressed by the compressor $\mathcal{C}$. We can summarize the basic EF mechanism in the following (for simplicity, we consider the deterministic and single client setup in the introduction):

$$\delta_t := \mathbf{g}_t - \mathbf{e}_t, \quad \hat{\mathbf{g}}_t := \mathcal{C}(\delta_t), \quad \mathbf{e}_{t+1} := \mathbf{e}_t + \hat{\mathbf{g}}_t - \mathbf{g}_t, \tag{1}$$

The basic (and essentially the only) tool that people have been using to analyze it is the virtual iteration framework (Mania et al., 2017), which has been the foundation of most of the theoretical works on EF since some of the first theoretical papers on EF (Stich et al., 2018). We consider the virtual iterate $\widetilde{\mathbf{x}}_t$, defined as:

$$\widetilde{\mathbf{x}}_t := \mathbf{x}_t + \frac{1}{\gamma}\mathbf{e}_t.$$

The key insight here is that $\mathbf{e}_t := \sum_{k=0}^{t-1}(\hat{\mathbf{g}}_k - \mathbf{g}_k)$, i.e. the accumulation of all the gradient errors, and the virtual iterate takes the true gradients as the update, i.e. $\widetilde{\mathbf{x}}_{t+1} = \widetilde{\mathbf{x}}_t - \frac{1}{\gamma}\mathbf{g}_t$, where again, $\mathbf{g}_t = \nabla f(\mathbf{x}_t)$. This enables the analysis to use the virtual iterate as a proxy for the gradient descent trajectories.

However, the combination of EF with virtual iteration does not extend directly to the composite setting. If we still construct $\hat{\mathbf{g}}_t$ by Equation (1) but update via a proximal step

$$\mathbf{x}_{t+1} = \underset{\mathbf{x} \in \text{dom}\psi}{\arg\min}\left\{h_t[\langle\hat{\mathbf{g}}_t, \mathbf{x} - \mathbf{x}_t\rangle + \psi(\mathbf{x})] + \frac{1}{2}\|\mathbf{x} - \mathbf{x}_t\|^2\right\}, \tag{2}$$

then the virtual iterate $\widetilde{\mathbf{x}}_t := \mathbf{x}_t - h_t\mathbf{e}_t$ is difficult to interpret, as it may lie outside $\text{dom}\,\psi$ and thus cannot serve as a feasible proxy.

To contrast, when $\psi \equiv 0$ the iterates satisfy

$$\mathbf{x}_t = \mathbf{x}_0 - \frac{1}{\gamma}\sum_{k=0}^{t-1}\hat{\mathbf{g}}_k = \mathbf{x}_0 - \frac{1}{\gamma}\left(\left(\sum_{k=0}^{t-1}\mathbf{g}_k\right) - \mathbf{e}_t\right),$$

so $\mathbf{x}_t$ is simply the cumulative sum of gradient estimates, and subtracting $\mathbf{e}_t$ recovers the exact gradient-descent trajectory. This additive structure is what makes the virtual iterate analysis effective.

When $\psi \not\equiv 0$, however, the proximal step in (2) introduces distortions at each iteration. The iterates $\mathbf{x}_t$ can no longer be expressed as a clean sum of past gradient estimates, while $\mathbf{e}_t$ remains a sum of compression errors. This structural mismatch is precisely why the classical virtual-iterate argument breaks down in the composite case.

## 1.2 Our Strategies

Following our discussions above, it is clear that the classical EF mechanism and the virtual iteration framework need to be modified in order to handle the composite setting. In particular, we need to restore the simple sum of gradient estimates in the iterates, so that $\mathbf{e}_t$ can still be used to correct the accumulated deviations from the true gradients. This reminds us of the *Dual Averaging* framework, where the algorithm sums up all the past gradients and take one step from the initial point at each step. In general, we consider the following update rule:

$$\mathbf{x}_{t+1} := \underset{\mathbf{x} \in \mathrm{dom}\psi}{\arg\min} \left\{ \sum_{k=0}^{t} a_k (\langle \hat{\mathbf{g}}_k, \mathbf{x} \rangle + \psi(\mathbf{x})) + \frac{\gamma_t}{2} \|\mathbf{x} - \mathbf{x}_0\|^2 \right\},$$

where $a_k, \gamma_t > 0$ are some properly chosen coefficients. In this way, the iterates $\mathbf{x}_t$ are defined precisely by the (weighted) sum of all gradient estimates $\sum_{k=0}^{t-1} a_k \hat{\mathbf{g}}_k$. We can therefore consider the (weighted) cumulative gradient error $\mathbf{e}_t := \sum_{k=0}^{t-1} a_k (\hat{\mathbf{g}}_k - \mathbf{g}_k)$ and use it to correct the deviations of $\mathbf{x}_t$ from the true gradient trajectory, this time inside the proximal operator:

$$\widetilde{\mathbf{x}}_{t+1} := \underset{\mathbf{x} \in \mathrm{dom}\psi}{\arg\min} \left\{ \sum_{k=0}^{t} a_k (\langle \hat{\mathbf{g}}_k, \mathbf{x} \rangle + \psi(\mathbf{x})) - \langle \mathbf{e}_t, \mathbf{x} \rangle + \frac{\gamma_t}{2} \|\mathbf{x} - \mathbf{x}_0\|^2 \right\}$$

$$= \underset{\mathbf{x} \in \mathrm{dom}\psi}{\arg\min} \left\{ \sum_{k=0}^{t} a_k (\langle \mathbf{g}_k, \mathbf{x} \rangle + \psi(\mathbf{x})) + \frac{\gamma_t}{2} \|\mathbf{x} - \mathbf{x}_0\|^2 \right\}.$$

It turns out that this intuitive modification of EF and the virtual iteration framework is precisely what we need to address the composite setting.

## 2 Problem Formulation and Assumptions

We consider the following distributed stochastic optimization problem:

$$F^* = \min_{\mathbf{x} \in \mathrm{dom}\psi} \left[ F(\mathbf{x}) = f(\mathbf{x}) + \psi(\mathbf{x}) \right], \qquad \text{where } f(\mathbf{x}) := \frac{1}{n} \sum_{i=1}^{n} f_i(\mathbf{x}), \tag{3}$$

where $\mathbf{x} \in \mathbb{R}^d$ are the parameters of a model that we train. We assume this problem has a solution which we denote by $\mathbf{x}^\star$. The objective function $F$ is a composite objective with the smooth part $f(\mathbf{x}) := \frac{1}{n} \sum_{i=1}^{n} f_i(\mathbf{x})$ and the composite part $\psi : \mathbb{R}^d \to \mathbb{R} \cup \{+\infty\}$. $\psi$ is a simple proper closed convex function. We write $\mathrm{dom}\psi \subset \mathbb{R}^d$ to be the set where $\psi$ is finite. Each function $f_i$ is a local loss function associated with a local data set $\mathcal{D}_i$, which can only be accessed by client $i$. There are in total $n$ clients indexed by $i \in \{1, \ldots, n\}$. The composite part $\psi$ can be accessed by the server.

Let us define the problem class that we consider in this paper. There are two types of agents in this problem: the server and the clients. The server has access to the proximal oracle for any $\mathbf{g}, \mathbf{x} \in \mathbb{R}^d$ and $\gamma \in \mathbb{R}_+$, defined as $\arg\min_{\mathbf{x}' \in \mathrm{dom}\psi} \left[ \langle \mathbf{g}, \mathbf{x}' \rangle + \psi(\mathbf{x}') + \frac{\gamma}{2} \|\mathbf{x} - \mathbf{x}'\|^2 \right]$. We assume that each client $i$ can access only the function $f_i$ and only via the stochastic gradient oracle as follows:

**Assumption 2.1.** For any $\mathbf{x} \in \mathrm{dom}\psi$, $\mathbf{g}_i(\mathbf{x}, \xi^i)$ is a stochastic gradient oracle for $f_i$ at $\mathbf{x}$, where $\xi^i$ is the randomness used by the oracle. We assume that $\mathbf{g}_i(\mathbf{x}, \xi^i)$ is unbiased and has bounded variance:

$$\mathbb{E} \left[ \mathbf{g}_i(\mathbf{x}, \xi^i) \right] = \nabla f_i(\mathbf{x}), \quad \mathbb{E}_{\xi^i} \left[ \|\mathbf{g}_i(\mathbf{x}, \xi^i) - \nabla f_i(\mathbf{x})\|^2 \right] \leq \sigma^2. \tag{4}$$

We consider the distributed setting where communication from the client to the server is expensive, and we need to compress it to reduce the cost. By (contractive) compression, we mean the following:

**Definition 2.2.** We say that a (possibly randomized) mapping $\mathcal{C}(\cdot, \zeta)\colon \mathbb{R}^d \to \mathbb{R}^d$ is a contractive compression operator if for some constant $0 < \delta \le 1$ it holds

$$\mathbb{E}_\zeta \left[ \|\mathcal{C}(\mathbf{s}, \zeta) - \mathbf{s}\|^2 \right] \le (1 - \delta)\|\mathbf{s}\|^2 \quad \forall \mathbf{s} \in \mathbb{R}^d. \tag{5}$$

Here $\zeta$ is some possible randomness used by the compressor. For simplicity, we will often omit $\zeta$ in the notation when there is no confusion.

In addition, we assume that the cost of communication from the server to each client is negligible (Karimireddy et al., 2019; Richtárik et al., 2021; Gao et al., 2024a), while the client can communicate with the server through the following two types of channels:

- **Compressed channel:** The client can send a compressed vector $\mathcal{C}(\mathbf{x}, \zeta) \in \mathbb{R}^d$ to the server, where $\mathcal{C}$ is a contractive compression operator (see Definition 2.2). The cost of sending one compressed vector is $1$.
- **Uncompressed channel:** The client can send a vector $\mathbf{g} \in \mathbb{R}^d$ to the server without any compression. The cost of sending one uncompressed vector is $m \ge 1$.

When the compressor is the Top-$K$ compressor (i.e. the client only sends the top $K$ elements of the gradient), then the cost of sending one uncompressed vector in $\mathbb{R}^d$ is at most $d/K$. In general, given any $\delta$-compression in the sense of Definition 2.2, we can combine at most $\mathcal{O}(\frac{1}{\delta} \log \frac{1}{\delta'})$ compressed messages to recover an $\delta'$-compression for any $\delta' > 0$ (He et al., 2023). In this sense, one can typically approximate an uncompressed channel with a compressed channel with an $\widetilde{\mathcal{O}}(\frac{1}{\delta})$ additional multiplicative overhead. That is, we can typically think of $m$ to be of the order $\frac{1}{\delta}$.

In this work, we are interested in minimizing the total (client to server, uplink) *communication cost* of the algorithm (for each client). Suppose that throughout the algorithm, each client makes $a$ compressed communications and $b$ uncompressed communications to the server, then the total communication cost is $a + mb$. This is roughly proportionate to $a + \frac{b}{\delta}$. We do not consider the communication cost from the server to the client (broadcast, downlink cost) since it is typically much lower than the uplink cost, which is conventional in prior works (Karimireddy et al., 2019; Richtárik et al., 2021; Gao et al., 2024a).

Let us now list the assumptions on the objective functions that we make in the paper. First, we make the standard assumption that $f$ is convex.

**Assumption 2.3.** We assume that the function $f$ and $\psi$ are convex, closed and proper over the convex domain $\mathrm{dom}\psi$.

We note that we do not assume that each local function $f_i$ is convex. We also assume that $f$ is $L$-smooth, which is standard in the literature (Stich et al., 2018; Karimireddy et al., 2019; Richtárik et al., 2021; Gao et al., 2024a).

**Assumption 2.4.** We assume that the objective function $f$ has $L$-Lipschitz gradients, i.e. for all $\mathbf{x}, \mathbf{y} \in \mathrm{dom}\psi$, it holds

$$\|\nabla f(\mathbf{x}) - \nabla f(\mathbf{y})\| \le L\|\mathbf{x} - \mathbf{y}\|. \tag{6}$$

We also assume the following smoothness condition for the local functions $f_i$.

**Assumption 2.5.** We assume that there exists some $\ell > 0$ such that for all $\mathbf{x}, \mathbf{y} \in \mathrm{dom}(\psi)$, it holds

$$\frac{1}{n} \sum_{i=1}^{n} \|\nabla f_i(\mathbf{x}) - \nabla f_i(\mathbf{y})\|^2 \le \ell^2 \|\mathbf{x} - \mathbf{y}\|^2. \tag{7}$$

*Remark* 2.6. Note that this is a weaker condition than what many existing works assume, e.g. (Richtárik et al., 2021; Li & Richtárik, 2021), where they assume that all $f_i$'s are $L_{\max}$-smooth. In contrast, we only require that they are in some sense smooth on average, which is strictly weaker.

We point out that by Jensen's inequality, we always have that $L \le \ell$. In the analysis of our main method, Algorithm 2, we eventually only need Assumption 2.5. However, Assumption 2.4 is still important for the analysis of the inexact dual averaging framework that we propose, as it does not presume any finite-sum structure of $f$.

---

**Algorithm 1** Inexact Dual Averaging

---

1: **Input:** $\mathbf{x}_0$ and $\{a_t, \gamma_t \in \mathbb{R}_+\}_{t=0,\dots,\infty}$. $\gamma_t$ is non-decreasing.
2: **for** $t = 0, 1, \dots$ **do**
3: $\quad$ | $\quad$ Obtain $\hat{\mathbf{g}}_t \approx \mathbf{g}_t \coloneqq \mathbf{g}(\mathbf{x}_t, \xi_t)$, $\xi_t$ is an independent copy of $\xi$.
4: $\quad$ | $\quad$ $\mathbf{x}_{t+1} = \arg\min_{\mathbf{x}} \left[ \Phi_t(\mathbf{x}) \coloneqq \sum_{k=0}^t a_k(f(\mathbf{x}_k) + \langle \hat{\mathbf{g}}_k, \mathbf{x} - \mathbf{x}_k \rangle + \psi(\mathbf{x})) + \frac{\gamma_t}{2} \|\mathbf{x} - \mathbf{x}_0\|^2 \right]$

---

## 3 THE INEXACT DUAL AVERAGING METHOD

In this section, we take a step back from the distributed optimization problem with communication compression that we consider in the rest of the paper, and consider solving a general stochastic composite optimization problem of the form $F^\star = \min_{\mathbf{x} \in \mathrm{dom}\psi}[F(\mathbf{x}) = f(\mathbf{x}) + \psi(\mathbf{x})]$. This perspective allows us to develop the core analytical tool that underpins our later analysis with compressed communication. Here, we do not assume that $f$ has a finite-sum structure. We make Assumptions 2.3 and 2.4 for the objective in this section. We assume that we have access to a stochastic gradient oracle $\mathbf{g}(\mathbf{x}, \xi)$ satisfying Assumption 3.1 below:

**Assumption 3.1.** For any $\mathbf{x} \in \mathrm{dom}\psi$, $\mathbf{g}(\mathbf{x}, \xi)$ is a stochastic gradient oracle for $f$ at $\mathbf{x}$. We assume that $\mathbf{g}(\mathbf{x}, \xi)$ is unbiased and has bounded variance:

$$\mathbb{E}[\mathbf{g}(\mathbf{x}, \xi)] = \nabla f(\mathbf{x}), \quad \mathbb{E}_\xi \left[ \|\mathbf{g}(\mathbf{x}, \xi) - \nabla f(\mathbf{x})\|^2 \right] \leq \sigma_{\mathbf{g}}^2. \tag{8}$$

We study the convergence of the general inexact dual averaging algorithm, as summarized in Algorithm 1, for solving this problem. The algorithm gets some inexact gradient $\hat{\mathbf{g}}_t$ that approximates the stochastic gradient $\mathbf{g}_t \coloneqq \mathbf{g}(\mathbf{x}_t, \xi_t)$ at each iteration. It uses these gradient estimates to perform a dual averaging update, with stepsize parameters $a_t$ and $\gamma_t$. We assume that $\gamma_t$ is non-decreasing.

We analyze the convergence of this method from the perspective of the virtual iterates, which are defined in Equation (9). We note that these virtual iterates are not explicitly computed or stored anywhere in the algorithm. However, since our convergence analysis will be given in terms of the suboptimality of a convex combination of or random sample of the virtual iterates, an immediate question would be how to output such a convex combination or random sample at the end of the algorithm without explicitly storing and computing the virtual iterates. We will addres this in Section 3.1.

Let's write $\bar{\mathbf{g}}_t \coloneqq \sum_{k=0}^t a_k \mathbf{g}_k$. We define the following virtual iteration, with $\widetilde{\mathbf{x}}_0 = \mathbf{x}_0$:

$$
\begin{aligned}
\widetilde{\mathbf{x}}_{t+1} &\coloneqq \arg\min_{\mathbf{x} \in \mathrm{dom}\psi} \left\{ \widetilde{\Phi}_t(\mathbf{x}) \coloneqq \sum_{k=0}^t a_k(f(\mathbf{x}_k) + \langle \mathbf{g}_k, \mathbf{x} - \mathbf{x}_k \rangle + \psi(\mathbf{x})) + \frac{\gamma_t}{2} \|\mathbf{x} - \mathbf{x}_0\|^2 \right\} \\
&= \arg\min_{\mathbf{x} \in \mathrm{dom}\psi} \left\{ \langle \bar{\mathbf{g}}_t, \mathbf{x} \rangle + \psi(\mathbf{x}) \sum_{k=0}^t a_k + \frac{\gamma_t}{2} \|\mathbf{x} - \mathbf{x}_0\|^2 \right\}.
\end{aligned}
\tag{9}
$$

Now, we define the accumulative error of the compressions:

$$\mathbf{e}_t \coloneqq \sum_{k=0}^{t-1} a_k(\hat{\mathbf{g}}_k - \mathbf{g}_k). \tag{10}$$

We first show that the distance between the virtual iterate $\widetilde{\mathbf{x}}_t$ and the actual iterate $\mathbf{x}_t$ is controlled by the accumulated error $\mathbf{e}_t$:

**Lemma 3.2.** *For any $t \geq 0$, we have:*

$$\|\widetilde{\mathbf{x}}_t - \mathbf{x}_t\|^2 \leq \frac{1}{\gamma_{t-1}^2} \|\mathbf{e}_t\|^2. \tag{11}$$

We simply write $\gamma_{-1} = \gamma_0$. Note that $\mathbf{e}_0 = \mathbf{0}$. With this, we can give the main convergence theorem for the *virtual iterates*:

**Theorem 3.3.** *Given Assumptions 3.1, 2.3 and 2.4 and $\gamma_{t-1} \geq 4a_t L$, then for any $\mathbf{x} \in \mathrm{dom}\psi$ and any $T \geq 1$, we have*

$$\sum_{t=0}^{T-1} \mathbb{E}\left[a_t(F(\widetilde{\mathbf{x}}_{t+1}) - F(\mathbf{x}))\right] + \frac{\gamma_{T-1}}{2}\mathbb{E}\left[\|\mathbf{x} - \widetilde{\mathbf{x}}_T\|^2\right]$$

$$\leq \frac{\gamma_{T-1}}{2}\|\mathbf{x} - \mathbf{x}_0\|^2 + L\sum_{t=0}^{T-1} \frac{a_t}{\gamma_{t-1}^2}\mathbb{E}\left[\|\mathbf{e}_t\|^2\right] + \sum_{t=0}^{T-1} \frac{a_t^2 \sigma_{\mathbf{g}}^2}{\gamma_{t-1}}. \quad (12)$$

*In addition, we have the following upper bound on the distance between consecutive iterates:*

$$\sum_{t=0}^{T-1}\left(\frac{\gamma_t + \gamma_{t-1} - a_t L}{2a_t}r_t^2 + \langle \hat{\mathbf{g}}_t - \nabla f(\mathbf{x}_t), \mathbf{x}_{t+1} - \mathbf{x}_t\rangle\right) \leq F_0 + \frac{1}{2}\sum_{t=0}^{T-1}(\beta_t \rho_t^2 - \beta_t \rho_{t+1}^2), \quad (13)$$

*where we write $\beta_t := \frac{\gamma_t - \gamma_{t-1}}{a_t}, \rho_t^2 := \|\mathbf{x}_t - \mathbf{x}_0\|^2, r_t^2 := \|\mathbf{x}_{t+1} - \mathbf{x}_t\|^2$ and $F_0 := F(\mathbf{x}_0) - F^\star$.*

Again, we note that Equation (12) deals with the virtual iterates. When $\psi \equiv 0$, typically we can bound the distance between $f(\widetilde{\mathbf{x}}_t)$ and $f(\mathbf{x}_t)$ simply by $\mathbb{E}\left[\|\mathbf{e}_t\|^2\right]$. This is, however, unclear when $\psi \not\equiv 0$. It is possible to directly analyze the behavior of $\mathbf{x}_t$ without using the virtual iterates at all, but the analysis obtained that way will be weaker due to the presence of $\psi$ (see Appendix I for a more detailed discussion, we further comment here that the techniques employed in Appendix I can also be used to obtain an analysis of the proximal method without dual averaging, albeit with similarly weak guarantees). It remains an open question whether it is possible to directly analyze $\mathbf{x}_t$ without using the virtual iterates and still obtain a result as strong as Theorem 3.3.

In addition, we also obtain an upper bound on the distance between $\mathbf{x}_{t+1}$ and $\mathbf{x}_t$, which will be useful later. Similar upper bounds on the distance between consecutive iterates have been used in many existing works that applied the gradient difference compression strategies (Richtárik et al., 2021; Fatkhullin et al., 2023; Gao et al., 2024a), but these are typically upper bounding the individual distances. Due to the dual averaging strategies, our analysis here is significantly different, and we are only able to upper bound the sum of the distances.

We point out that controlling the error $\|\hat{\mathbf{g}}_t - \nabla f(\mathbf{x}_t)\|^2$ is method-dependent, that is, it depends on how we constructed the approximate $\hat{\mathbf{g}}_t$. Therefore we do not further analyze this term here, and we discuss this term in more details when we present the analysis of our main algorithm in this work.

### 3.1 A SAMPLING PROCEDURE FOR THE VIRTUAL ITERATES

Provided that the errors are sufficiently small, Theorem 3.3 allows us to establish the convergence rate in terms of $\frac{1}{A_T}\sum_{t=0}^{T-1} a_t[F(\widetilde{\mathbf{x}}_{t+1}) - F^\star]$, where $\widetilde{\mathbf{x}}_t$ are the virtual iterates rather than the real iterates $\mathbf{x}_t$ and we write $A_t = \sum_{s=0}^{t-1} a_s$.

Therefore, after $T$ steps, we would like to return a randomly chosen point among $\{\widetilde{\mathbf{x}}_1, \ldots, \widetilde{\mathbf{x}}_T\}$ with the probabilities proportional to $a_t$. This can be implemented as follows: at each iteration $t$, we keep track of the accumulated true gradients $\bar{\mathbf{g}}_t = \sum_{s=0}^t a_s \mathbf{g}_s$ and update $\bar{\mathbf{g}}$ to $\bar{\mathbf{g}}_t$ when $\tau_t = 1$, and it remains unchanged when $\tau_t = 0$, where $\tau_t$ is a Bernoulli variable with probability $\Pr[\tau = 1] = \frac{a_t}{A_{t+1}}$. This way, at step $T - 1$, $\bar{\mathbf{g}}$ is a random sample from the set $\{\bar{\mathbf{g}}_t\}_{t\in\{0,\cdots,T-1\}}$ with probabilities proportional to $a_t$,. see Proposition E.1. Using $\bar{\mathbf{g}}$, we can easily compute a random sample $\bar{\mathbf{x}}_T$ from the set $\{\widetilde{\mathbf{x}}_t\}_{t=1,\ldots,T}$ as follows:

$$\bar{\mathbf{x}}_T = \arg\min_{\mathbf{x}\in\mathrm{dom}\psi}\left[\langle \bar{\mathbf{g}}, \mathbf{x}\rangle + \psi(\mathbf{x})\sum_{t=0}^{T-1} a_t + \frac{\gamma_\tau}{2}\|\mathbf{x} - \mathbf{x}_0\|^2\right].$$

We summarize this procedure in Algorithm 3 in Appendix E. As a consequence, we have the following:

**Lemma 3.4.** *The output $\bar{\mathbf{x}}_T$ from Algorithm 3 is a random variable over the set $\{\widetilde{\mathbf{x}}_t\}_{t\in[T]}$, where $\widetilde{\mathbf{x}}_t$ is defined in Equation* (9). *In particular, we have for any $\mathbf{x} \in \mathrm{dom}(\psi)$ (that are independent of $\{\xi_t, \tau_t\}_{t\in[T-1]}$):*

$$\mathbb{E}_{\tau_0,\ldots,\tau_{T-1},\xi_0,\ldots,\xi_{T-1}}\left[F(\bar{\mathbf{x}}_T) - F(\mathbf{x})\right] = \frac{1}{A_T}\sum_{t=0}^{T-1} a_t \mathbb{E}_{\xi_0,\ldots,\xi_{T-1}}\left[F(\widetilde{\mathbf{x}}_{t+1}) - F(\mathbf{x})\right]. \quad (14)$$

---

**Algorithm 2** EControl with Dual Averaging

---

1: **Input:** $\mathbf{x}_0, \eta, \mathbf{e}_0^i = \mathbf{0}, \hat{\mathbf{g}}_{-1}^i = \nabla f_i(\mathbf{x}_0, \xi_0^i)$.
2: **for** $t = 0, 1, \ldots$ **do**
3:     **Server:**
4:     Sample $\tau_t = 1$ with prob. $\frac{1}{t+1}$ and $\tau_t = 0$ otherwise. Send $\tau_t$ to all clients.
5:     **clients:**
6:     $\mathbf{g}_t^i = \nabla f_i(\mathbf{x}_t, \xi_t^i)$ where $\xi_t^i$ is independent copy of $\xi^i$.    $\bar{\mathbf{g}}_t^i = \bar{\mathbf{g}}_t^i + \mathbf{g}_t^i$
7:     $\bar{\mathbf{g}}^i = \bar{\mathbf{g}}_t^i$ if $\tau_t = 1$ otherwise $\bar{\mathbf{g}}^i$ remains.
8:     $\delta_t^i = \mathbf{g}_t^i - \hat{\mathbf{g}}_{t-1}^i - \eta \mathbf{e}_t^i, \Delta_t^i = \mathcal{C}(\delta_t^i, \zeta_t^i)$   where $\zeta_t^i$ is independent copy of $\zeta^i$.
9:     $\hat{\mathbf{g}}_t^i = \hat{\mathbf{g}}_{t-1}^i + \Delta_t^i, \; \mathbf{e}_{t+1}^i = \mathbf{e}_t^i + \hat{\mathbf{g}}_t^i - \mathbf{g}_t^i$
10:    send $\Delta_t^i$ to the server
11:    **server**
12:    $\hat{\mathbf{g}}_t = \hat{\mathbf{g}}_{t-1} + \frac{1}{n}\sum_{i=1}^n \Delta_t^i$
13:    $\mathbf{x}_{t+1} = \arg\min_{\mathbf{x}}\{\Phi_t(\mathbf{x}) := \sum_{s=0}^t (f(\mathbf{x}_s) + \langle \hat{\mathbf{g}}_s, \mathbf{x} - \mathbf{x}_s \rangle + \psi(\mathbf{x})) + \frac{\gamma_t}{2}\|\mathbf{x} - \mathbf{x}_0\|^2\}$
14: **client:** send $\bar{\mathbf{g}}^i$ to the server
15: **server:**
16: $\bar{\mathbf{g}} = \frac{1}{n}\sum_{i=1}^n \bar{\mathbf{g}}^i$
17: $\bar{\mathbf{x}}_T = \arg\min_{\mathbf{x}}\left\{\langle \bar{\mathbf{g}}, \mathbf{x} \rangle + (\tau+1)\psi(\mathbf{x}) + \frac{\gamma_\tau}{2}\|\mathbf{x} - \mathbf{x}_0\|^2\right\}$ where $\tau$ is the last $t$ s.t. $\tau_t = 1$.

---

# 4 EControl with Dual Averaging

In this section, we apply the general framework discussed in Section 3 to the particular case of distributed optimization with communication compression. In such a setting, the stochastic gradient in Assumption 3.1 is the average of the stochastic gradient of each client $i$, which follows Assumption 2.1. Therefore, $\sigma_{\mathbf{g}}^2 = \frac{\sigma^2}{n}$ where $n$ is the number of clients. Now the gradient estimate $\hat{\mathbf{g}}_t$ is the average of $\hat{\mathbf{g}}_t^i$ where each $\hat{\mathbf{g}}_t^i$ is each client's estimate of its local gradient $\mathbf{g}_t^i := \mathbf{g}(\mathbf{x}_t, \xi_t^i)$, which can be communicated to the server using compressed communication channels.

The sampling procedure in Section 3.1 can be easily implemented in such a setting. The variables $\bar{\mathbf{g}}_t$ and $\bar{\mathbf{g}}$ do not need to be maintained and communicated by the server throughout the algorithm; instead, we can simply ask the workers to maintain their local $\bar{\mathbf{g}}_t^i$ and $\bar{\mathbf{g}}^i$, using the same random bit $\tau_t$ (which costs 1 bit of communication). At the end of the algorithm, we use one full communication round to collect the local $\bar{\mathbf{g}}^i$ and compute the output $\bar{\mathbf{x}}_T$. In total, the above procedure costs exactly 1 round of full communication plus one extra bit in each of the $T$ communication rounds.

Now, as the main focus of this section, we present a specific mechanism of generating the $\hat{\mathbf{g}}_t \approx \mathbf{g}_t$, the EControl method (summarized in Algorithm 4 in Appendix G), using mainly compressed communication channels. We assume that $a_t = 1$ for all $t$. For simplicity, in this section we also assume that $\gamma_t = \gamma$ for all $t$ for some constant $\gamma > 0$. In Appendix H, we present a more advanced analysis of Algorithm 4 that handles variable $\gamma_t$. The variable stepsize analysis for EControl mechanism is unknown prior to this work due to the complexity of $\eta$ parameter in EControl and we have to employ a scaling/rescaling strategy in the analysis to handle it. We slightly modified the presentation from (Gao et al., 2024a) to suit our setup better. We can put Algorithms 1, 3 and 4 together to get our final algorithm, EControl with Dual Averaging, summarized in Algorithm 2 (see Appendix F for a more detailed walk-through of the algorithm). We highlight the EControl module with green color.

We note that the EControl mechanism was first proposed in (Gao et al., 2024a) and analyzed under the condition $\psi \equiv 0$. In Appendix G we briefly discuss some intuitions behind the design of EControl. Here we present a more systematic and hopefully cleaner analysis. We simply bound the sum of errors by the average of stochastic gradient differences. We note that the following upper bounds are entirely the consequences of the EControl mechanism, independent of the specific properties of the objectives and oracles.

**Lemma 4.1.** *Let $\eta = \frac{\delta}{3\sqrt{1-\delta}(1+\sqrt{1-\delta})}$, then:*

$$\sum_{t=1}^{T}\|\mathbf{e}_t^i\|^2 \leq \frac{81(1-\delta)^2(1+\sqrt{1-\delta})^4}{2\delta^4}\sum_{t=0}^{T-2}\|\mathbf{g}_{t+1}^i - \mathbf{g}_t^i\|^2,$$

$$\sum_{t=0}^{T-1}\|\hat{\mathbf{g}}_t^i - \mathbf{g}_t^i\|^2 \leq \frac{36(1-\delta)(1+\sqrt{1-\delta})^2}{\delta^2}\sum_{t=0}^{T-2}\|\mathbf{g}_{t+1}^i - \mathbf{g}_t^i\|^2. \tag{15}$$

*Remark* 4.2. We point out that in the analysis of the classical EF mechanism, upper bounding $\frac{1}{n}\sum_{i=1}^{n}\|\mathbf{e}_t^i\|^2$ relies on upper bounding $\frac{1}{n}\sum_{i=1}^{n}\|\nabla f_i(\mathbf{x}_t)\|^2$, which leads to the data heterogeneity assumption, but more importantly, requires upper bounds on $\|\nabla f(\mathbf{x}_t)\|^2$ in terms of the function residuals. When $\psi \equiv 0$, this follows directly from the smoothness of $f$. However, in the composite setting, this is no longer possible unless $\nabla f(\mathbf{x}^\star) = \mathbf{0}$, which is not true in general. In contrast, EControl uses the gradient difference compression technique to obtain a better handle on the errors and we only need to upper bound $\frac{1}{n}\sum_{i=1}^{n}\|\mathbf{g}_{t+1}^i - \mathbf{g}_t^i\|^2$, which again can be done via Assumption 2.5 and Equation (13).

Next, we invoke the specific properties regarding the smooth objective $f$ and the stochastic oracles, and apply Theorem 3.3 to get an upper bound on the sum of $\|\mathbf{g}_{t+1}^i - \mathbf{g}_t^i\|^2$, and consequently, an upper bound on the sum of $\|\mathbf{e}_t^i\|^2$.

**Lemma 4.3.** *Given Assumptions 2.1 and 2.3 to 2.5, and let $\eta = \frac{\delta}{3\sqrt{1-\delta}(1+\sqrt{1-\delta})}, \gamma \geq \frac{24\sqrt{2}\ell}{\delta}$, then we have:*

$$\sum_{t=0}^{T-1}\frac{1}{n}\sum_{i=1}^{n}\mathbb{E}\left[\|\mathbf{g}_{t+1}^i - \mathbf{g}_t^i\|^2\right] \leq \frac{80\ell^2}{9\gamma}F_0 + 7T\sigma^2. \tag{16}$$

*Therefore, by Lemma 4.1, we also have:*

$$\sum_{t=0}^{T-1}\frac{1}{n}\sum_{i=1}^{n}\mathbb{E}\left[\|\mathbf{e}_t^i\|^2\right] \leq \frac{5760\ell^2}{\delta^4\gamma}F_0 + \frac{4536T\sigma^2}{\delta^4}. \tag{17}$$

Finally, combining all of the pieces in Sections 3, 3.1 and 4, we can give the overall convergence guarantee of our final Algorithm 2.

**Theorem 4.4.** *Given Assumptions 2.3 to 2.5, and setting $a_t = 1, \gamma_T = \gamma, \eta = \frac{\delta}{3\sqrt{1-\delta}(1+\sqrt{1-\delta})}$, and taking one initial stochastic gradient step from $\mathbf{x}_0$ to $\mathbf{x}_0'$ if $\psi \not\equiv 0$ and setting*

$$\gamma = \max\left\{\frac{24\sqrt{2}\ell}{\delta}, \sqrt{\frac{T\sigma^2}{nR_0^2}}, \frac{17T^{1/3}\ell^{1/3}\sigma^{2/3}}{R_0^{2/3}\delta^{4/3}}\right\},$$

*then it takes at most*

$$T = \frac{16R_0^2\sigma^2}{n\varepsilon^2} + \frac{561R_0^2\sqrt{\ell}\sigma}{\delta^2\varepsilon^{3/2}} + \frac{96\sqrt{2}\ell R_0^2}{\delta\varepsilon},$$

*iterations of Algorithm 2 to get $\mathbb{E}\left[F(\bar{\mathbf{x}}_T) - F^\star\right] \leq \varepsilon$. Here, $R_0 := \|\mathbf{x}_0 - \mathbf{x}^\star\|$.*

*In particular, this means that with three rounds of uncompressed communication (one for the initial stochastic gradient step, one communicating $\hat{\mathbf{g}}_{-1}$ and one communicating $\bar{\mathbf{g}}$), and $T$ rounds of compressed communications, Algorithm 2 reaches the desired accuracy. Assuming that the cost of sending compressed vectors is $1$ and sending uncompressed vectors is $m$, it costs at most*

$$\frac{16R_0^2\sigma^2}{n\varepsilon^2} + \frac{561R_0^2\sqrt{\ell}\sigma}{\delta^2\varepsilon^{3/2}} + \frac{96\sqrt{2}\ell R_0^2}{\delta\varepsilon} + 3m,$$

*in communications for Algorithm 2 to get:*

$$\mathbb{E}\left[F(\bar{\mathbf{x}}_T) - F^\star\right] \leq \varepsilon.$$

*Remark* 4.5. In the statement of Theorem 4.4 we let the algorithm take one initial exact stochastic gradient step in the composite setting. This comes from the fact that we need one gradient step in the composite setting to upper bound $F_0$ by $LR_0^2$ (see Lemma C.3). This is satisfied automatically in the classical unconstrained setting. Without the extra initial step, the algorithm would still converge (with properly chosen step size), but the rate would additionally depend on $F_0$ (though it would still be a desirable $\mathcal{O}(\frac{1}{\delta\varepsilon})$ term). We refer to Appendix G for more details. We note that the rate in Theorem 4.4 matches the rate of EControl (Gao et al., 2024a) in the basic uncomposite setting when $\psi \equiv 0$.

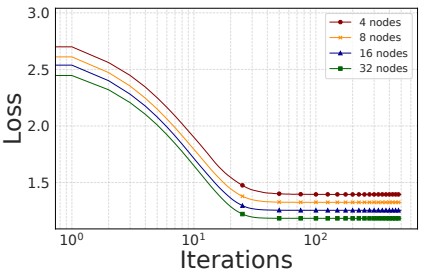 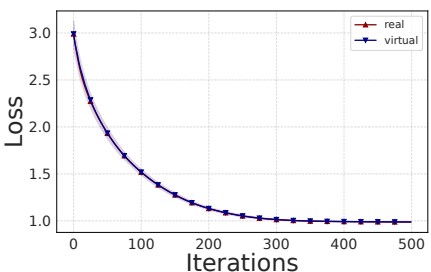

(a) **Achieving linear speedup.** Performance of EControl with Dual Averaging with increasing number of clients $n$. We fix $\gamma$ to be 0.0001. The error that the algorithm stabilizes around decreases as $n$ increases.

(b) **Virtual Iterates vs Real Iterates.** The performance of the virtual and real iterates of EControl with Dual Averaging. We see that the virtual iterates and real iterates perform similarly.

Figure 1: Performance of EControl with Dual Averaging on a synthetic regularized softmax objective.

## 5 EXPERIMENTS

In this section, we present some experimental results on a synthetic softmax objective with $\ell_1$ regularization to complement our theoretical analysis. Details of the experimental setup (including data generation) and an additional experiment on the FashionMNIST dataset are presented in Appendix B. All our codes for the experiments can be found at this link[1].

The softmax objective with $\ell_1$ regularization is given as: $\min_{\mathbf{x} \in \mathbb{R}^d} \left\{ F(\mathbf{x}) := \mu \log \left( \sum_{i=1}^k \exp \left[ \frac{\langle a_i, \mathbf{x} \rangle - b_i}{\mu} \right] \right) + \lambda \|\mathbf{x}\|_1 \right\}$, where $\mu$ controls the smoothness, and we set it to $\mu = 0.1$. We set the regularization parameter $\lambda = 0.1$. We set the dimension $d = 200$ and the total number of samples $k = 2048$. We simulate the stochastic gradient by adding Gaussian noise to the gradients. We use Top-K compressor with $K/d = 0.1$. For both of the following experiments, we set $\sigma^2 = 25$.

**Linear speedup with $n$:** one of the key characteristics of EF-style algorithms is that the leading (stochastic) term in its rate improves linearly with the number of clients $n$ and is $\delta$-free. We prove that EControl with Dual Averaging does satisfy this quality—with the catch that the theory only applies to (the random sample of) virtual iterates. Here we verify this property experimentally for the *real iterates* directly. We fix a small enough $\gamma$ to be 0.0001, and increase the number of clients $n$. The results are summarized in Figure 1a. We see that the error that real iterates stabilize around decreases linearly with $n$, verifying the linear speedup for *real iterates* as well.

**Virtual iterates vs real iterates:** while we can do the sampling procedure to obtain convergence in terms of the virtual iterates, this is ultimately still somewhat clumsy in practice. The real iterates, on the other hand, do not enjoy convergence bounds that are as good. Here, we compare the suboptimality of the virtual and the real iterates. The results are summarized in Figure 1b. We see that the virtual and real iterates perform almost identically in the suboptimality. This suggests that the real iterates might also be amenable to a strong theory; future work might explore refining our analytical template in Appendix I to achieve this, or construct lower bound examples to demonstrate a gap between the virtual and real iterates.

## 6 CONCLUSION

In this work, we addressed the open challenge of combining error feedback with composite optimization. We showed that the classical virtual-iterate approach breaks down in this setting, as the composite update destroys the additive structure that underpins its analysis. To resolve this, we introduced the first framework that integrates error feedback with dual averaging, which restores the summation structure and enables control of accumulated compression errors. Our analysis extends

---

[1]https://github.com/mlolab/comp-opt-da_iclr

the theory of error feedback to the convex composite case and recovers the best-known results in the unconstrained setting when $\psi \equiv 0$.

Looking ahead, our inexact dual averaging analysis provides a versatile template for problems where iterative updates are distorted by approximation, noise, or constraints. This opens up promising directions in domains such as safe reinforcement learning, constrained distributed optimization, and large-scale learning under resource limitations. An exciting avenue for future work is to connect our approach with recent efforts that aim to simplify error-feedback methods for practical use, potentially leading to more robust and scalable communication-efficient algorithms.

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

## CONTENTS

## A  Related Works on Error Feedback and Communication Compression

In this section, we survey some of the most relevant works on EF. We note that while there's a rich body of literature on EF in the uncomposite setting, the extension to the composite setting is less developed. Stich et al. (2018); Alistarh et al. (2018); Karimireddy et al. (2019) were among the first to explore the theoretical properties of the practical EF mechanism proposed by Seide et al. (2014), but their analyses are restricted to the single-client setting. Under certain forms of bounded data heterogeneity assumption (e.g. bounded gradient, bounded gradient dissimilarity, or bounded local objective gap at optimum), Cordonnier (2018); Alistarh et al. (2018); Stich & Karimireddy (2020) extended the analysis to the more realistic multi-client settings. But these data heterogeneity assumptions are indeed very limiting factors. These theories were further refined in (Beznosikov et al., 2020; Stich, 2020).

Another line of work parallel to the classic EF variants is the gradient difference compression mechanism. Mishchenko et al. (2019) added an additional *unbiased* compressor for gradient difference into the EF framework to address the issue of data heterogeneity and obtained the DIANA algorithm. Another of follow-up works include Gorbunov et al. (2020); Stich (2020); Qian et al. (2021b), and culminated in the EF21 algorithm (Richtárik et al., 2021). The EF21 algorithm is the the first to fully support contractive compression in the full gradient regime. However, it is not compatible with stochastic gradients and leads to non-convergence up to the variance of the stochastic oracle. This was later addressed by adding momentum in (Fatkhullin et al., 2023), or by a more careful blend of EF and gradient difference compression in (Gao et al., 2024a). The latter work proposed the EControl mechanism, which is the basis of Algorithm 4 in this paper.

All of the above focus on the uncomposite setting, and their extensions to the composite setting remain largely unexplored. Qian et al. (2020) considered a proximal variant of the EF mechanism, which they called EC-ProxSGD. However, their work considered the finite-sum stochastic setting, and their convergence rates have $\mathcal{O}(\frac{1}{\delta^2})$ dependence on the compression quality, which is suboptimal compared to the state-of-the-art EF variants in the uncomposite setting. Perhaps more relevant to our work is their EC-RDA algorithm, which is a dual averaging variant of the basic EF mechanism in the finite-sum setting. However, the analysis of EC-RDA relies on (in addition to smoothness) a number of bounded gradient assumptions on the objectives and the regularizers which we do not assume in our work, and their rates have a cubic dependence on $\frac{1}{\delta}$ which is even more undesirable. In (Qian et al., 2021a), the authors proposed and analyzed several variant-reduced EF algorithms in the composite setting, but these algorithms are designed for the finite-sum setting and are not applicable to our setting. In particular, their EC-LSVRG requires periodic access to full gradients. Their EC-Quartz and EC-SDCA considers a more specific form of composite objectives and requires access to the first order information of the conjugates of the regularizers. More recently, Islamov et al. (2025) analyzed a variant of EF, called Safe-EF, when $\psi$ is an indicator function of some convex set $Q$. Their analysis requires that the constraint set $Q$ be described as an intersection of sublevel sets of functions, with first-order information of these functions available. Under this structural assumption, their method blends updates in the direction of both the objective and the constraint functions, enabling the virtual iterate to account for constraints. Safe-EF also assumes that the stochastic gradients are bounded, which circumvents the issue of upper bounding $\|\nabla f(\mathbf{x}_t)\|^2$ in the smooth case.

Since Richtárik et al. (2021) first analyzed EF21 in the non-composite full gradient regime, there have been some attempts to extend EF21 to the composite setting. In particular, Fatkhullin et al. (2025) analyzed a proximal version of EF21, but only in the nonconvex and full gradient regime. Condat et al. (2022) proposed the EF-BV algorithm, which (in addition to unifying EF21 with DIANA) extends the analysis of EF21 to the composite setting. But EF-BV's analysis is also restricted to the full gradient regime, and assumes either the PL or KL condition with strictly positive constants. It is unclear whether their analysis can be extended to the general convex setting even with full gradients. Recently, Islamov et al. (2025) extended the analysis of EF21 to the general convex composite setting, but noted that their analysis requires a bounded domain assumption, which we do not assume in our work.

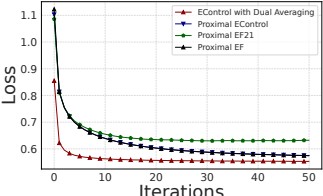 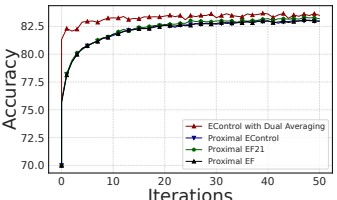

Figure 2: **Superior performance** Comparison of the performance of EControl with Dual Averaging, proximal EF, and proximal EF21 on the FashionMNIST classification problem with $\ell_1$ regularization. We use Top-$K$ compression with $\delta = 0.1$. We see that EControl with Dual Averaging significantly outperforms the other methods.

## B  ADDITIONAL EXPERIMENTS AND DETAILS

In this section, we provide some additional experimental details for our experiments in Section 5, and an additional experiment on the FashionMNIST dataset.

### B.1  SYNTHETIC SOFTMAX OBJECTIVE

We generate the data $\{\mathbf{a}_i, b_i\}$ randomly, following Moshtaghifar et al. (2024): we generate i.i.d. vectors $\hat{\mathbf{a}}_i$ whose entries are sampled from $[-1, 1]$ uniformly at random. Each $b_i$ is generated the same way. This leads to a preliminary objective $\hat{f}$. We then set $\mathbf{a}_i := \hat{\mathbf{a}}_i - \nabla \hat{f}(\mathbf{0})$. The resulting $\{\mathbf{a}_i, b_i\}$ gives us the desired objective $f$ with $\mathbf{0}$ being the minimizer.

For the experiment comparing the virtual and the real iterates, we perform a grid search for the stepsize parameters over $\frac{1}{\gamma} \in \{0.1, 0.05, 0.01, 0.005, 0.001, 0.0005, 0.0001\}$.

### B.2  REGULARIZED FASHIONMNIST CLASSIFICATION

We now consider a logistic regression problem with $\ell_1$ regularization on the FashionMNIST dataset (Xiao et al., 2017). We set the regularization parameter $\lambda = 0.001$. We compare the performance of EControl with Dual Averaging against the proximal EF and the proximal EF21 methods. Following our synthetic experiments, we choose to evaluate the performance of EControl with Dual Averaging directly with the real iterates. We split the FashionMNIST dataset into $n = 10$ clients, and distribute half of the dataset randomly to each client, and assign the rest of the dataset according to their labels, i.e. data with label $i$ is distributed to client $i$. We use Top-K compressor with $K/d = 0.1$. We use batch size $64$. We perform a grid search for the stepsize parameters over $\frac{1}{\gamma} \in \{0.1, 0.01, 0.001, 0.0001\}$. The results are summarized in Figure 2. We see that EControl with Dual Averaging significantly outperforms the other methods. In additional, we note that EControl with Dual Averaging admits a much larger stepsize than the other methods, which might explain its superior performance.

## C  AUXILIARY FACTS AND RESULTS

In this section we collect some auxiliary facts and results that are useful for the analysis of our algorithms. The first one is a simple fact regarding the square of the norm of a sum of vectors.

**Fact C.1.** For any $\gamma_1, \ldots, \gamma_T$, we have:

$$\|\sum_{t=1}^{T} \gamma_t\|^2 \leq T \sum_{t=1}^{T} \|\gamma_t\|^2. \tag{18}$$

The next lemma upper bounds an exponentially weighted sum of positive sequences:

**Lemma C.2.** *Given a sequence of non-negative values $\{\alpha_t\}_{t \in [T-1]}$, and some other sequences $\{u_t\}_{t \in [T-1]}$. If there exists $\gamma \in (0, 1)$ such that the following holds:*

$$\alpha_{t+1} \leq (1 - \beta)\alpha_t + u_t, \quad \alpha_0 = 0, \tag{19}$$

*then we have:*

$$\alpha_{t+1}^2 \le \frac{1}{\beta} \sum_{k=0}^{t} (1-\beta)^{t-k} u_k^2, \quad \sum_{t=0}^{T} \alpha_{t+1}^2 \le \frac{1}{\beta^2} \sum_{t=0}^{T} u_t^2. \tag{20}$$

*Proof.* Since $\alpha_0 = 0$, we have:

$$\alpha_{t+1} \le \sum_{k=0}^{t} (1-\beta)^{t-k} u_k.$$

Squaring both sides, and applying Jensen's inequality, we have:

$$\alpha_{t+1}^2 \le \frac{1}{S_t} \sum_{k=0}^{t} (1-\beta)^{t-k} u_k^2,$$

where $S_t := \sum_{k=0}^{t} (1-\beta)^{t-k}$. It's easy to check that $S_t \le \frac{1}{\beta}$, and therefore we get the first part of Equation (20). Now summing this from $t = 0$ to $T$, we get:

$$\sum_{t=0}^{T} \alpha_{t+1}^2 \le \sum_{k=0}^{T} \left( \sum_{t=k}^{T} (1-\beta)^{t-k} \right) u_k^2,$$

Note that $\sum_{t=k}^{T} (1-\beta)^{t-k} \le \frac{1}{\beta}$, and therefore we get the second part of Equation (20). $\square$

Now we show that one gradient step will lead to an upper bound on the objective value.

**Lemma C.3.** *Let $f$ be convex and $L$-smooth, and $\mathbf{x}_0 \in \mathrm{dom}(\psi)$ and $\mathbf{g}_0$ satisfying Assumption 3.1, consider $\mathbf{x}_0'$ defined as the following:*

$$\mathbf{x}_0' := \arg\min \left[ f(\mathbf{x}_0) + \langle \mathbf{g}_0, \mathbf{x} - \mathbf{x}_0 \rangle + \psi(\mathbf{x}) + \frac{\gamma_0}{2} \|\mathbf{x} - \mathbf{x}_0\|^2 \right].$$

*then for any $\mathbf{y} \in \mathrm{dom}\psi$ and $\|\mathbf{y} - \mathbf{x}_0\|^2 \le R^2$, if we choose $\gamma_0 := \max\{2L, \frac{\sqrt{2}\sigma}{R}\}$, we have:*

$$\mathbb{E}\left[F(\mathbf{x}_0') - F(\mathbf{y})\right] \le LR^2 + \frac{R\sigma}{\sqrt{2}}, \quad \mathbb{E}\left[\|\mathbf{y} - \mathbf{x}_0'\|^2\right] \le 2R^2. \tag{21}$$

*Proof.*

$$f(\mathbf{x}_0) + \langle \mathbf{g}_0, \mathbf{x}_0' - \mathbf{x}_0 \rangle + \psi(\mathbf{x}_0') + \frac{\gamma_0}{2} \|\mathbf{x}_0' - \mathbf{x}_0\|^2 + \frac{\gamma_0}{2} \|\mathbf{y} - \mathbf{x}_0'\|^2$$

$$\le f(\mathbf{x}_0) + \langle \mathbf{g}_0, \mathbf{y} - \mathbf{x}_0 \rangle + \psi(\mathbf{y}) + \frac{\gamma_0}{2} \|\mathbf{y} - \mathbf{x}_0\|^2$$

$$= f(\mathbf{x}_0) + \langle \nabla f(\mathbf{x}_0), \mathbf{y} - \mathbf{x}_0 \rangle + \psi(\mathbf{y}) + \frac{\gamma_0}{2} \|\mathbf{y} - \mathbf{x}_0\|^2 + \langle \mathbf{g}_0 - \nabla f(\mathbf{x}_0), \mathbf{y} - \mathbf{x}_0 \rangle$$

$$\le F(\mathbf{y}) + \frac{1}{2} \|\mathbf{y} - \mathbf{x}_0\|^2 + \langle \mathbf{g}_0 - \nabla f(\mathbf{x}_0), \mathbf{y} - \mathbf{x}_0 \rangle.$$

On the other hand, we have:

$$f(\mathbf{x}_0) + \langle \mathbf{g}_0, \mathbf{x}_0' - \mathbf{x}_0 \rangle + \psi(\mathbf{x}_0') + \frac{\gamma_0}{2} \|\mathbf{x}_0' - \mathbf{x}_0\|^2 + \frac{\gamma_0}{2} \|\mathbf{y} - \mathbf{x}_0'\|^2$$

$$= f(\mathbf{x}_0) + \langle \nabla f(\mathbf{x}_0), \mathbf{x}_0' - \mathbf{x}_0 \rangle + \psi(\mathbf{x}_0') + \langle \mathbf{g}_0 - \nabla f(\mathbf{x}_0), \mathbf{x}_0' - \mathbf{x}_0 \rangle + \frac{\gamma_0}{2} \|\mathbf{x}_0' - \mathbf{x}_0\|^2 + \frac{\gamma_0}{2} \|\mathbf{y} - \mathbf{x}_0'\|^2$$

$$\ge F(\mathbf{x}_0') + \langle \mathbf{g}_0 - \nabla f(\mathbf{x}_0), \mathbf{x}_0' - \mathbf{x}_0 \rangle + \frac{\gamma_0 - L}{2} \|\mathbf{x}_0' - \mathbf{x}_0\|^2 + \frac{\gamma_0}{2} \|\mathbf{y} - \mathbf{x}_0'\|^2$$

$$\ge F(\mathbf{x}_0') - \frac{1}{\gamma_0} \|\mathbf{g}_0 - \nabla f(\mathbf{x}_0)\|^2 + \frac{\gamma_0 - 2L}{4} \|\mathbf{x}_0' - \mathbf{x}_0\|^2 + \frac{\gamma_0}{2} \|\mathbf{y} - \mathbf{x}_0'\|^2.$$

Putting these together, we have:

$$F(\mathbf{x}_0') - F(\mathbf{y}) + \frac{\gamma_0}{2} \|\mathbf{y} - \mathbf{x}_0'\|^2 + \frac{\gamma_0 - 2L}{4} \|\mathbf{x}_0' - \mathbf{x}_0\|^2 \le \frac{\gamma_0}{2} \|\mathbf{y} - \mathbf{x}_0\|^2 + \frac{1}{\gamma_0} \|\mathbf{g}_0 - \nabla f(\mathbf{x}_0)\|^2 + \langle \mathbf{g}_0 - \nabla f(\mathbf{x}_0), \mathbf{y} - \mathbf{x}_0 \rangle.$$

Now by Assumption 3.1, we take the expectation and get:

$$\mathbb{E}\left[\frac{1}{\gamma_0}\|\mathbf{g}_0 - \nabla f(\mathbf{x}_0)\|^2 + \langle \mathbf{g}_0 - \nabla f(\mathbf{x}_0), \mathbf{y} - \mathbf{x}_0\rangle\right] \leq \frac{\sigma^2}{\gamma_0}.$$

Therefore, assuming that $\gamma_0 \geq 2L$, we have:

$$\mathbb{E}\left[F(\mathbf{x}_0') - F(\mathbf{y})\right] + \frac{\gamma_0}{2}\mathbb{E}\left[\|\mathbf{y} - \mathbf{x}_0'\|^2\right] \leq \frac{\gamma_0}{2}R^2 + \frac{\sigma^2}{\gamma_0}.$$

Now we pick $\gamma_0 = \max\{2L, \frac{\sqrt{2}\sigma}{R}\}$, then we have:

$$\mathbb{E}\left[F(\mathbf{x}_0') - F(\mathbf{y})\right] \leq LR^2 + \frac{R\sigma}{\sqrt{2}}.$$

In addition, we have:

$$\mathbb{E}\left[\|\mathbf{y} - \mathbf{x}_0'\|^2\right] \leq R^2 + 2\gamma_0^2\sigma^2 \leq 2R^2.$$

$\square$

## D   ANALYSIS OF INEXACT DUAL AVERAGING

In this section we give the missing proofs for the analysis of Algorithm 1. We first introduce the following notation:

$$\widetilde{\Phi}_t^\star := \widetilde{\Phi}_t(\widetilde{\mathbf{x}}_{t+1}), \quad \Phi_t^\star := \Phi_t(\mathbf{x}_{t+1}),$$

the optimum of the virtual and real subproblems at $t$.

We now present the proof of Lemma 3.2:

**Lemma 3.2.** *For any $t \geq 0$, we have:*

$$\|\widetilde{\mathbf{x}}_t - \mathbf{x}_t\|^2 \leq \frac{1}{\gamma_{t-1}^2}\|\mathbf{e}_t\|^2. \tag{11}$$

*Proof.* By the definition of $\widetilde{\Phi}_t$, $\Phi_t$ and $\mathbf{x}_t$, we have:

$$\mathbf{x}_{t+1} = \underset{\mathbf{x} \in \mathrm{dom}(\psi)}{\arg\min} \left\{\widetilde{\Phi}_t(\mathbf{x}) + \langle \mathbf{e}_{t+1}, \mathbf{x}\rangle\right\}.$$

Therefore, we have for any $\mathbf{x} \in \mathrm{dom}(\psi)$:

$$\widetilde{\Phi}_t(\mathbf{x}) + \langle \mathbf{e}_{t+1}, \mathbf{x}\rangle \geq \widetilde{\Phi}_t(\mathbf{x}_{t+1}) + \langle \mathbf{e}_{t+1}, \mathbf{x}\rangle + \frac{\gamma_t}{2}\|\mathbf{x} - \mathbf{x}_{t+1}\|^2$$

$$\geq \widetilde{\Phi}_t^\star + \langle \mathbf{e}_{t+1}, \mathbf{x}_t\rangle + \frac{\gamma_t}{2}\|\mathbf{x} - \mathbf{x}_{t+1}\|^2 + \frac{\gamma_t}{2}\|\widetilde{\mathbf{x}}_{t+1} - \mathbf{x}_{t+1}\|^2.$$

Now plug in the choice $\mathbf{x} := \widetilde{\mathbf{x}}_{t+1}$, we have:

$$\langle \mathbf{e}_{t+1}, \widetilde{\mathbf{x}}_{t+1} - \mathbf{x}_{t+1}\rangle \geq \gamma_t\|\widetilde{\mathbf{x}}_{t+1} - \mathbf{x}_{t+1}\|^2.$$

Note that we have $\langle \mathbf{e}_{t+1}, \widetilde{\mathbf{x}}_{t+1} - \mathbf{x}_{t+1}\rangle \leq \|\mathbf{e}_{t+1}\|\|\widetilde{\mathbf{x}}_{t+1} - \mathbf{x}_{t+1}\|$, we get the desired result. $\square$

We now present the proof for Theorem 3.3:

**Theorem 3.3.** *Given Assumptions 3.1, 2.3 and 2.4 and $\gamma_{t-1} \geq 4a_t L$, then for any $\mathbf{x} \in \mathrm{dom}\psi$ and any $T \geq 1$, we have*

$$\sum_{t=0}^{T-1} \mathbb{E}\left[a_t(F(\widetilde{\mathbf{x}}_{t+1}) - F(\mathbf{x}))\right] + \frac{\gamma_{T-1}}{2}\mathbb{E}\left[\|\mathbf{x} - \widetilde{\mathbf{x}}_T\|^2\right]$$

$$\leq \frac{\gamma_{T-1}}{2}\|\mathbf{x} - \mathbf{x}_0\|^2 + L\sum_{t=0}^{T-1}\frac{a_t}{\gamma_{t-1}^2}\mathbb{E}\left[\|\mathbf{e}_t\|^2\right] + \sum_{t=0}^{T-1}\frac{a_t^2\sigma_\mathbf{g}^2}{\gamma_{t-1}}. \tag{12}$$

*In addition, we have the following upper bound on the distance between consecutive iterates:*

$$\sum_{t=0}^{T-1}\left(\frac{\gamma_t + \gamma_{t-1} - a_t L}{2a_t}r_t^2 + \langle \hat{\mathbf{g}}_t - \nabla f(\mathbf{x}_t), \mathbf{x}_{t+1} - \mathbf{x}_t\rangle\right) \leq F_0 + \frac{1}{2}\sum_{t=0}^{T-1}(\beta_t\rho_t^2 - \beta_t\rho_{t+1}^2), \tag{13}$$

*where we write $\beta_t := \frac{\gamma_t - \gamma_{t-1}}{a_t}, \rho_t^2 := \|\mathbf{x}_t - \mathbf{x}_0\|^2, r_t^2 := \|\mathbf{x}_{t+1} - \mathbf{x}_t\|^2$ and $F_0 := F(\mathbf{x}_0) - F^\star$.*

*Proof.* By the definition of $\widetilde{\mathbf{x}}_t$, we have for any $\mathbf{x} \in \mathrm{dom}\psi$:

$$\widetilde{\Phi}_t(\mathbf{x}) \geq \widetilde{\Phi}_t^\star + \frac{\gamma_t}{2}\|\mathbf{x} - \widetilde{\mathbf{x}}_{t+1}\|^2.$$

We also have:

$$\widetilde{\Phi}_t(\mathbf{x}) = \sum_{k=0}^t a_k(f(\mathbf{x}_k) + \langle \nabla f(\mathbf{x}_k), \mathbf{x} - \mathbf{x}_k \rangle + \psi(\mathbf{x})) + \sum_{k=0}^t a_k \langle \mathbf{g}_k - \nabla f(\mathbf{x}_k), \mathbf{x} - \mathbf{x}_k \rangle + \frac{\gamma_t}{2}\|\mathbf{x} - \mathbf{x}_0\|^2$$

$$\overset{(i)}{\leq} \sum_{k=0}^t a_k F(\mathbf{x}) + \gamma \sum_{k=0}^t a_k \langle \mathbf{g}_k - \nabla f(\mathbf{x}_k), \mathbf{x} - \mathbf{x}_k \rangle + \frac{\gamma_t}{2}\|\mathbf{x} - \mathbf{x}_0\|^2,$$

where in $(i)$ we used the convexity of $f$. Note that the gradient noise $\mathbf{g}_k - \nabla f(\mathbf{x}_k)$ is independent of $\mathbf{x} - \mathbf{x}_k$ for fixed $\mathbf{x}$ independent of the algorithm (in particular, for $\mathbf{x}^\star$). Therefore, taking expectation on both sides:

$$\mathbb{E}_{\xi_0,\ldots,\xi}\left[\widetilde{\Phi}_t(\mathbf{x})\right] \leq \sum_{k=0}^t a_k F(\mathbf{x}) + \frac{\gamma_t}{2}\|\mathbf{x} - \mathbf{x}_0\|^2.$$

Now by the definition of $\widetilde{\mathbf{x}}_t$, we have:

$$\widetilde{\Phi}_t^\star = \widetilde{\Phi}_{t-1}(\widetilde{\mathbf{x}}_{t+1}) + a_t(f(\mathbf{x}_t) + \langle \nabla f(\mathbf{x}_t), \widetilde{\mathbf{x}}_{t+1} - \mathbf{x}_t \rangle + \psi(\widetilde{\mathbf{x}}_{t+1}))$$

$$+ a_t \langle \mathbf{g}_t - \nabla f(\mathbf{x}_t), \widetilde{\mathbf{x}}_{t+1} - \mathbf{x}_t \rangle + \frac{\gamma_t - \gamma_{t-1}}{2}\|\widetilde{\mathbf{x}}_{t+1} - \mathbf{x}_0\|$$

$$\overset{(ii)}{\geq} \widetilde{\Phi}_{t-1}^\star + \frac{\gamma_{t-1}}{2}\|\widetilde{\mathbf{x}}_{t+1} - \widetilde{\mathbf{x}}_t\|^2 + a_t(f(\mathbf{x}_t) + \langle \nabla f(\mathbf{x}_t), \widetilde{\mathbf{x}}_{t+1} - \mathbf{x}_t \rangle + \psi(\widetilde{\mathbf{x}}_{t+1}))$$

$$+ + a_t \langle \mathbf{g}_t - \nabla f(\mathbf{x}_t), \widetilde{\mathbf{x}}_{t+1} - \mathbf{x}_t \rangle + \frac{\gamma_t - \gamma_{t-1}}{2}\|\widetilde{\mathbf{x}}_{t+1} - \mathbf{x}_0\|$$

$$\overset{(iii)}{\geq} \widetilde{\Phi}_{t-1}^\star + \frac{\gamma_{t-1}}{2}\|\widetilde{\mathbf{x}}_{t+1} - \widetilde{\mathbf{x}}_t\|^2 + a_t(f(\widetilde{\mathbf{x}}_{t+1}) + \psi(\widetilde{\mathbf{x}}_{t+1}) - \frac{L}{2}\|\widetilde{\mathbf{x}}_{t+1} - \mathbf{x}_t\|^2)$$

$$+ a_t \langle \mathbf{g}_t - \nabla f(\mathbf{x}_t), \widetilde{\mathbf{x}}_{t+1} - \mathbf{x}_t \rangle + \frac{\gamma_t - \gamma_{t-1}}{2}\|\widetilde{\mathbf{x}}_{t+1} - \mathbf{x}_0\|$$

$$\overset{(iv)}{\geq} \widetilde{\Phi}_{t-1}^\star + \frac{\gamma_{t-1}}{2}\|\widetilde{\mathbf{x}}_{t+1} - \widetilde{\mathbf{x}}_t\|^2 + a_t(F(\widetilde{\mathbf{x}}_{t+1}) - L\|\widetilde{\mathbf{x}}_{t+1} - \widetilde{\mathbf{x}}_t\|^2 - L\|\widetilde{\mathbf{x}}_t - \mathbf{x}_t\|^2)$$

$$+ a_t \langle \mathbf{g}_t - \nabla f(\mathbf{x}_t), \widetilde{\mathbf{x}}_{t+1} - \mathbf{x}_t \rangle + \frac{\gamma_t - \gamma_{t-1}}{2}\|\widetilde{\mathbf{x}}_{t+1} - \mathbf{x}_0\|$$

$$\overset{(v)}{\geq} \widetilde{\Phi}_{t-1}^\star + \frac{\gamma_{t-1} - 2a_t L}{2}\|\widetilde{\mathbf{x}}_{t+1} - \widetilde{\mathbf{x}}_t\|^2 + a_t F(\widetilde{\mathbf{x}}_{t+1}) - a_t L\|\widetilde{\mathbf{x}}_t - \mathbf{x}_t\|^2$$

$$+ a_t \langle \mathbf{g}_t - \nabla f(\mathbf{x}_t), \widetilde{\mathbf{x}}_{t+1} - \mathbf{x}_t \rangle,$$

where in $(ii)$ we used the strong convexity of $\widetilde{\Phi}_t$ and in $(iii)$ we used Assumption 2.4 . In $(iv)$ we used Young's inequality and in $(v)$ we used that assumption that $\gamma_t$ is non-decreasing.Note that the gradient noise $\mathbf{g}_t - \nabla f(\mathbf{x}_t)$ is independent of $\mathbf{x}_t$ and $\widetilde{\mathbf{x}}_t$, we have:

$$\mathbb{E}_{\xi_t}\left[\widetilde{\Phi}_t^\star | \xi_0, \ldots, \xi_{t-1}\right] \geq \mathbb{E}_{\xi_t}\left[\widetilde{\Phi}_{t-1}^\star + \frac{\gamma_{t-1} - 2a_t L}{2}\|\widetilde{\mathbf{x}}_{t+1} - \widetilde{\mathbf{x}}_t\|^2 + a_t F(\widetilde{\mathbf{x}}_{t+1}) - a_t L\|\widetilde{\mathbf{x}}_t - \mathbf{x}_t\|^2 | \xi_0, \ldots, \xi_{t-1}\right]$$

$$+ a_t \mathbb{E}_{\xi_t}\left[\langle \mathbf{g}_t - \nabla f(\mathbf{x}_t), \widetilde{\mathbf{x}}_{t+1} - \widetilde{\mathbf{x}}_t \rangle | \xi_0, \ldots, \xi_{t-1}\right]$$

$$\geq \mathbb{E}_{\xi_t}\left[\widetilde{\Phi}_{t-1}^\star + \frac{\gamma_{t-1} - 4a_t L}{4}\|\widetilde{\mathbf{x}}_{t+1} - \widetilde{\mathbf{x}}_t\|^2 + a_t F(\widetilde{\mathbf{x}}_{t+1}) - a_t L\|\widetilde{\mathbf{x}}_t - \mathbf{x}_t\|^2 | \xi_0, \ldots, \xi_{t-1}\right]$$

$$- \frac{a_t^2 \sigma_{\mathbf{g}}^2}{\gamma_{t-1}}.$$

Now rearranging and summing from $t = 0$ to $T - 1$, and using the law of total expectation, we get:

$$\sum_{t=0}^{T-1} a_t \mathbb{E}_{\xi_0,\dots,\xi_{T-1}} \left[ F(\widetilde{\mathbf{x}}_{t+1}) + \frac{\gamma_{t-1} - 4a_t L}{4} \|\widetilde{\mathbf{x}}_{t+1} - \widetilde{\mathbf{x}}_t\|^2 \right]$$

$$\leq \mathbb{E}_{\xi_0,\dots,\xi_{T-1}} \left[ \widetilde{\Phi}_{T-1}^\star \right] + L \sum_{t=0}^{T-1} a_t \mathbb{E}_{\xi_0,\dots,\xi_{T-1}} \left[ \|\widetilde{\mathbf{x}}_t - \mathbf{x}_t\|^2 \right] + \sum_{t=0}^{T-1} \frac{a_t^2 \sigma_{\mathbf{g}}^2}{\gamma_{t-1}}$$

$$\leq \mathbb{E}_{\xi_0,\dots,\xi_{T-1}} \left[ \widetilde{\Phi}_{T-1}(\mathbf{x}) \right] - \frac{\gamma_{T-1}}{2} \mathbb{E}_{\xi_0,\dots,\xi_{T-1}} \left[ \|\mathbf{x} - \widetilde{\mathbf{x}}_T\|^2 \right]$$

$$+ L \sum_{t=0}^{T-1} a_t \mathbb{E}_{\xi_0,\dots,\xi_{T-1}} \left[ \|\widetilde{\mathbf{x}}_t - \mathbf{x}_t\|^2 \right] + \sum_{t=0}^{T-1} \frac{a_t^2 \sigma_{\mathbf{g}}^2}{\gamma_{t-1}}$$

$$\leq \sum_{s=0}^{T-1} a_t F(\mathbf{x}) + \frac{\gamma_{T-1}}{2} \|\mathbf{x} - \mathbf{x}_0\|^2 - \frac{\gamma_{T-1}}{2} \mathbb{E}_{\xi_0,\dots,\xi_{T-1}} \left[ \|\mathbf{x} - \widetilde{\mathbf{x}}_T\|^2 \right]$$

$$+ L \sum_{t=0}^{T-1} a_t \mathbb{E}_{\xi_0,\dots,\xi_{T-1}} \left[ \|\widetilde{\mathbf{x}}_t - \mathbf{x}_t\|^2 \right] + \sum_{t=0}^{T-1} \frac{a_t^2 \sigma_{\mathbf{g}}^2}{\gamma_{t-1}}.$$

Rearranging, we get the desired result.

$$\sum_{t=0}^{T-1} \mathbb{E}_{\xi_0,\dots,\xi_{T-1}} \left[ a_t(F(\widetilde{\mathbf{x}}_{t+1}) - F(\mathbf{x})) + \frac{\gamma_{t-1} - 4a_t L}{4} \|\widetilde{\mathbf{x}}_{t+1} - \widetilde{\mathbf{x}}_t\|^2 \right] + \frac{\gamma_{T-1}}{2} \mathbb{E}_{\xi_0,\dots,\xi_{T-1}} \left[ \|\mathbf{x} - \widetilde{\mathbf{x}}_T\|^2 \right]$$

$$\leq \frac{\gamma_{T-1}}{2} \|\mathbf{x} - \mathbf{x}_0\|^2 + L \sum_{t=0}^{T-1} a_t \mathbb{E}_{\xi_0,\dots,\xi_{T-1}} \left[ \|\widetilde{\mathbf{x}}_t - \mathbf{x}_t\|^2 \right] + \sum_{t=0}^{T-1} \frac{a_t^2 \sigma_{\mathbf{g}}^2}{\gamma_{t-1}}.$$

For Equation (13), by definition of $\mathbf{x}_{t+1}$, we have:

$$\Phi_t^\star = \Phi_{t-1}(\mathbf{x}_{t+1}) + a_t(f(\mathbf{x}_t) + \langle \hat{\mathbf{g}}_t, \mathbf{x}_{t+1} - \mathbf{x}_t \rangle + \psi(\mathbf{x}_{t+1})) + \frac{\gamma_t - \gamma_{t-1}}{2} \|\mathbf{x}_{t+1} - \mathbf{x}_0\|^2$$

$$\overset{(vi)}{\geq} \Phi_{t-1}^\star + \frac{\gamma_{t-1}}{2} \|\mathbf{x}_{t+1} - \mathbf{x}_t\|^2 + a_t(f(\mathbf{x}_t) + \langle \hat{\mathbf{g}}_t, \mathbf{x}_{t+1} - \mathbf{x}_t \rangle + \psi(\mathbf{x}_{t+1})) + \frac{\gamma_t - \gamma_{t-1}}{2} \|\mathbf{x}_{t+1} - \mathbf{x}_0\|^2$$

$$= \Phi_{t-1}^\star + \frac{\gamma_{t-1}}{2} \|\mathbf{x}_{t+1} - \mathbf{x}_t\|^2 + a_t(f(\mathbf{x}_t) + \langle \nabla f(\mathbf{x}_t), \mathbf{x}_{t+1} - \mathbf{x}_t \rangle + \langle \hat{\mathbf{g}}_t - \nabla f(\mathbf{x}_t), \mathbf{x}_{t+1} - \mathbf{x}_t \rangle + \psi(\mathbf{x}_{t+1}))$$

$$+ \frac{\gamma_t - \gamma_{t-1}}{2} \|\mathbf{x}_{t+1} - \mathbf{x}_0\|^2$$

$$\overset{(vii)}{\geq} \Phi_{t-1}^\star + \frac{\gamma_{t-1} - a_t L}{2} \|\mathbf{x}_{t+1} - \mathbf{x}_t\|^2 + a_t(F(\mathbf{x}_{t+1}) + \langle \hat{\mathbf{g}}_t - \nabla f(\mathbf{x}_t), \mathbf{x}_{t+1} - \mathbf{x}_t \rangle) + \frac{\gamma_t - \gamma_{t-1}}{2} \|\mathbf{x}_{t+1} - \mathbf{x}_0\|^2,$$

where in $(vi)$ we used the strong convexity of $\Phi_t$ and in $(ii)$ we used Assumption 2.4.

Again by the definition of $\mathbf{x}_{t+1}$, we have:

$$\Phi_t^\star + \frac{\gamma_t}{2} \|\mathbf{x}_{t+1} - \mathbf{x}_t\|^2 \overset{(viii)}{\leq} \Phi_t(\mathbf{x}_t)$$

$$= \Phi_{t-1}^\star + a_t F(\mathbf{x}_t) + \frac{\gamma_t - \gamma_{t-1}}{2} \|\mathbf{x}_t - \mathbf{x}_0\|^2,$$

where in $(viii)$ we used the strong convexity of $\Phi_{t+1}$.

Putting these together, we have:

$$\frac{\gamma_t + \gamma_{t-1} - a_t L}{2} \|\mathbf{x}_{t+1} - \mathbf{x}_t\|^2 + a_t(F(\mathbf{x}_{t+1}) + \langle \hat{\mathbf{g}}_t - \nabla f(\mathbf{x}_t), \mathbf{x}_{t+1} - \mathbf{x}_t \rangle) + \frac{\gamma_t - \gamma_{t-1}}{2} \|\mathbf{x}_{t+1} - \mathbf{x}_0\|^2$$

$$\leq a_t F(\mathbf{x}_t) + \frac{\gamma_t - \gamma_{t-1}}{2} \|\mathbf{x}_t - \mathbf{x}_0\|^2.$$

Now divide both sides by $a_t$ and sum from $t = 0$ to $T - 1$, we have:

$$\sum_{t=0}^{T-1} \left( \frac{\gamma_t + \gamma_{t-1} - a_t L}{2a_t} \|\mathbf{x}_{t+1} - \mathbf{x}_t\|^2 + \langle \hat{\mathbf{g}}_t - \nabla f(\mathbf{x}_t), \mathbf{x}_{t+1} - \mathbf{x}_t \rangle \right) \leq F(\mathbf{x}_0) - F(\mathbf{x}_T) + \frac{1}{2} \sum_{t=0}^{T-1} (\beta_t \rho_t^2 - \beta_t \rho_{t+1}^2).$$

$\square$

## E  SAMPLING PROCEDURE FOR VIRTUAL ITERATES

---

**Algorithm 3** Sampling Procedure for Virtual Iterates

---

1: $\bar{\mathbf{g}}, \bar{\mathbf{g}}_0 = \mathbf{0}$
2: **for** $t = 0, 1, \ldots$ **do**
3: $\quad A_t = \sum_{s=0}^{t-1} a_s$
4: $\quad$ Sample $\tau_t = 1$ with prob. $\frac{a_t}{A_{t+1}}$ and $\tau_t = 0$ otherwise.
5: $\quad$ Obtain $\hat{\mathbf{g}}_t \approx \mathbf{g}_t := \mathbf{g}(\mathbf{x}_t, \xi_t)$
6: $\quad \bar{\mathbf{g}}_t = \bar{\mathbf{g}}_{t-1} + a_t \mathbf{g}_t$
7: $\quad \bar{\mathbf{g}} = \bar{\mathbf{g}}_t$ if $\tau = 1$ otherwise $\bar{\mathbf{g}}$ remains.
8: $\quad$ Update $\mathbf{x}_{t+1}$
9: $\bar{\mathbf{x}}_T = \arg\min_{\mathbf{x}} \left[ \langle \bar{\mathbf{g}}, \mathbf{x} \rangle + \psi(\mathbf{x}) A_{\tau+1} + \frac{\gamma_\tau}{2} \|\mathbf{x} - \mathbf{x}_0\|^2 \right]$ where $\tau$ is the last $t$ such that $\tau_t = 1$.
10: $A_{\tau+1} := \sum_{t=0}^{\tau} a_t$

---

In this section we prove the missing results for the sampling procedure for the virtual iterates in Section 3.1. We first summarize the procedure for clarity as Algorithm 3.

Now a simple proposition regarding the sampling procedure. This is folklore knowledge and the proof is taken directly from (Gao et al., 2024b).

**Proposition E.1.** *Given a stream of points $\{\mathbf{x}_k\}_{k=1}^{\infty}$ in $\mathbb{R}^d$ and positive scalars $\{h_k\}_{k=1}^{\infty}$, we can maintain, at each step $k \geq 1$, the random variable $\mathbf{x}_{t(k)}$, where $t(k)$ is a random index from $\{1, \ldots, k\}$ chosen with probabilities $\Pr(t(k) = i) = \frac{h_i}{H_k}$, $i = 1, \ldots, k$, where $H_k := \sum_{i=1}^{k} h_i$. This requires only $\mathcal{O}(d)$ memory and computation.*

*Proof.* We maintain the variables $\bar{\mathbf{x}}_k \in \mathbb{R}^d$ and $H_k \in \mathbb{R}$ which are both initialized to 0 at step $k = 0$. Then, at each step $k \geq 1$, we update $H_k \leftarrow H_{k-1} + h_k$ and also, with probability $\frac{h_k}{H_k}$, we update $\bar{\mathbf{x}}_k \leftarrow \mathbf{x}_k$ (or, with probability $1 - \frac{h_k}{H_k}$, keep the old $\bar{\mathbf{x}}_k = \bar{\mathbf{x}}_{k-1}$). The memory and computation costs are $\mathcal{O}(d)$. Note that, for any $1 \leq i \leq k$, the event $\bar{\mathbf{x}}_k = \mathbf{x}_i$ happens iff $\bar{\mathbf{x}}$ was updated at step $i$ and then not updated at each step $j = i + 1, \ldots, k$. Hence, for any $1 \leq i \leq k$, we have

$$\Pr(\bar{\mathbf{x}}_k = \mathbf{x}_i) = \frac{h_i}{H_i} \cdot \prod_{j=i+1}^{k} \left( 1 - \frac{h_j}{H_j} \right) = \frac{h_i}{H_i} \cdot \prod_{j=i+1}^{k} \frac{H_{j-1}}{H_j} = \frac{h_i}{H_k}. \qquad \square$$

## F  DESCRIPTION OF FULL ALGORITHM

In this section, we describe Algorithm 2 in more details for clarity. The algorithm combines Algorithms 1 and 3 and Algorithm 4 together.

At each iteration, the server samples a bernoulli random variable $\tau_t$ to decide whether to update the $\bar{\mathbf{g}}^i$ vector, the cumulative gradient sample for all clients. The clients then proceed to compute their local stochastic gradient $\mathbf{g}_t^i$, and add it to their local cumulative gradient $\bar{\mathbf{g}}_t^i$. If $\tau_t = 1$, the client updates its cumulative gradient sample $\bar{\mathbf{g}}^i$ to $\bar{\mathbf{g}}_t^i$, otherwise it remains unchanged. Then the client make the EControl update, where it updates the local error $\mathbf{e}_{t+1}^i$ and the local gradient estimate $\hat{\mathbf{g}}_t^i$. The client then sends the compressed local gradient difference $\Delta_t^i$ to the server. Now the server collects the gradient differences $\Delta_t^i$ from all clients and updates the global gradient estimate $\hat{\mathbf{g}}_t$ and makes a dual averaging update to the primal variable $\mathbf{x}_{t+1}$.

Finally, the server collects the cumulative gradient samples $\bar{\mathbf{g}}^i$ from all clients via a full communication and computes $\bar{\mathbf{g}}$. The final output is then computed using $\bar{\mathbf{g}}$ so that it becomes a random sample of the virtual iterates (which are not explicitly computed and stored).

# G   ANALYSIS OF THE EControl MECHANISM

---
**Algorithm 4 EControl**

---
1: **Input:** $\mathbf{x}_0, \eta, \mathbf{e}_0^i = \mathbf{0}, \hat{\mathbf{g}}_{-1}^i = \nabla f_i(\mathbf{x}_0, \xi_0^i)$.
2: **for** $t = 0, 1, \ldots$ **do**
3:     **clients:**
4:     $\mathbf{g}_t^i = \mathbf{g}_i(\mathbf{x}_t, \xi_t^i)$, $\xi_t^i$ is independent copy of $\xi^i$
5:     $\delta_t^i = \mathbf{g}_t^i - \hat{\mathbf{g}}_{t-1}^i - \eta \mathbf{e}_t^i, \Delta_t^i = \mathcal{C}(\delta_t^i)$
6:     $\hat{\mathbf{g}}_t^i = \hat{\mathbf{g}}_{t-1}^i + \Delta_t^i$
7:     $\mathbf{e}_{t+1}^i = \mathbf{e}_t^i + \hat{\mathbf{g}}_t^i - \mathbf{g}_t^i$
8:     send $\Delta_t^i$ to the server
9:     **server**
10:     $\Delta_t = \frac{1}{n} \sum_{i=1}^n \Delta_t^i$
11:     $\hat{\mathbf{g}}_t = \hat{\mathbf{g}}_{t-1} + \Delta_t$

---

In this section we present the missing proofs for the analysis of Algorithm 2. For ease of understanding, we also summarize thet EControl mechanism in Algorithm 4.

The EControl mechanism is a blend of two different techniques. The first is the classical EF mechanism, which keeps track of the (local) compression errors $\mathbf{e}_t^i$ and feedbacks them to the compressor. The second is the gradient difference compression technique, which compresses the difference between the current gradient and the previous estimates. As was discussed in Gao et al. (2024a), directly mixing the two methods might lead to suboptimal dependence on the compression quality $\delta$ in the convergence rate. The key innovation of EControl is to introduce a scaling factor $\eta$ on the error feedback term. Note that the historical estimates $\hat{\mathbf{g}}_t^i$ also carries some information on the error, and the error feedback term should be scaled down to balance the two sources of information. The specific choice of $\eta$, as we explain below, is carefully chosen to optimize the dependence on $\delta$ in the final convergence rate.

Again, we remind the readers that for now we restrict ourselves to the setting where $a_t = 1$ and $\gamma_t = \gamma$. Please refer to Appendix H for more details on the case where $\gamma_t$ is changing (and non-decreasing).

We first present an upper bound on each sums of $\|\mathbf{e}_t^i\|^2$ and $\|\hat{\mathbf{g}}_t^i - \mathbf{g}_t^i\|^2$, both in terms of the sum of $\|\mathbf{g}_{t+1}^i - \mathbf{g}_t^i\|^2$.

**Lemma 4.1.** *Let* $\eta = \frac{\delta}{3\sqrt{1-\delta}(1+\sqrt{1-\delta})}$, *then:*

$$\sum_{t=1}^T \|\mathbf{e}_t^i\|^2 \leq \frac{81(1-\delta)^2(1+\sqrt{1-\delta})^4}{2\delta^4} \sum_{t=0}^{T-2} \|\mathbf{g}_{t+1}^i - \mathbf{g}_t^i\|^2,$$

$$\sum_{t=0}^{T-1} \|\hat{\mathbf{g}}_t^i - \mathbf{g}_t^i\|^2 \leq \frac{36(1-\delta)(1+\sqrt{1-\delta})^2}{\delta^2} \sum_{t=0}^{T-2} \|\mathbf{g}_{t+1}^i - \mathbf{g}_t^i\|^2.$$

$$(15)$$

*Proof.* By the definition of $\mathbf{e}_{t+1}^i$, we have:

$$\mathbf{e}_{t+1}^i := \hat{\mathbf{g}}_t^i - \mathbf{g}_t^i + \mathbf{e}_t^i = \hat{\mathbf{g}}_{t-1}^i + \Delta_t^i - \mathbf{g}_t^i + \mathbf{e}_t^i = \Delta_t^i - \delta_t^i + (1-\eta)\mathbf{e}_t^i,$$

Therefore, by triangular inequality, we have:

$$\|\mathbf{e}_{t+1}^i\| \leq (1-\eta)\|\mathbf{e}_t^i\| + \|\Delta_t^i - \delta_t^i\| \leq (1-\eta)\|\mathbf{e}_t^i\| + \sqrt{1-\delta}\|\delta_t^i\|,$$

where in the last inequality we used the definition of the compressor. Now by Lemma C.2, we get:

$$\sum_{t=1}^T \|\mathbf{e}_t^i\|^2 \leq \frac{1-\delta}{\eta^2} \sum_{t=0}^{T-1} \|\delta_t^i\|^2.$$

Next we note the following:

$$\begin{aligned}
\delta_{t+1}^i &= \mathbf{g}_{t+1}^i - \hat{\mathbf{g}}_t^i - \eta \mathbf{e}_{t+1}^i \\
&= \mathbf{g}_t^i - \hat{\mathbf{g}}_t^i - \eta(\hat{\mathbf{g}}_t^i - \mathbf{g}_t^i + \mathbf{e}_t^i) + \mathbf{g}_{t+1}^i - \mathbf{g}_t^i \\
&= (1+\eta)(\mathbf{g}_t^i - \hat{\mathbf{g}}_t^i) - \eta \mathbf{e}_t^i + \mathbf{g}_{t+1}^i - \mathbf{g}_t^i \\
&= (1+\eta)(\delta_t^i - \Delta_t^i + \eta \mathbf{e}_t^i) - \eta \mathbf{e}_t^i + \mathbf{g}_{t+1}^i - \mathbf{g}_t^i \\
&= (1+\eta)(\delta_t^i - \Delta_t^i) + \eta^2 \mathbf{e}_t^i + \mathbf{g}_{t+1}^i - \mathbf{g}_t^i.
\end{aligned}$$

Similar as before, we now apply triangular inequality and definition of the compressor and get:

$$\|\delta_{t+1}^i\| \leq (1+\eta)\sqrt{1-\delta}\|\delta_t^i\| + \eta^2 \|\mathbf{e}_t^i\| + \|\mathbf{g}_{t+1}^i - \mathbf{g}_t^i\|.$$

Let's write $\beta \equiv 1 - (1+\eta)\sqrt{1-\delta}$. Now we apply Lemma C.2 again and Young's inequality, and note that $\delta_0^i = \mathbf{0}$, we get:

$$\sum_{t=0}^{T-1} \|\delta_t^i\|^2 \leq \frac{2}{\beta^2} \sum_{t=0}^{T-2}(\eta^4 \|\mathbf{e}_t^i\|^2 + \|\mathbf{g}_{t+1}^i - \mathbf{g}_t^i\|^2).$$

Now we plug in the upper bound on the sum of $\|\mathbf{e}_t^i\|$ (and note that $\mathbf{e}_0^i = \mathbf{0}$):

$$\sum_{t=0}^{T-1} \|\delta_t^i\|^2 \leq \frac{2(1-\delta)\eta^2}{\beta^2} \sum_{t=0}^{T-3}\|\delta_t^i\|^2 + \frac{2}{\beta^2}\sum_{t=0}^{T-2}\|\mathbf{g}_{t+1}^i - \mathbf{g}_t^i\|^2.$$

Rearranging, we have:

$$\sum_{t=0}^{T-1} \|\delta_t^i\|^2 \leq \frac{2}{\beta^2 - 2(1-\delta)\eta^2} \sum_{t=0}^{T-2}\|\mathbf{g}_{t+1}^i - \mathbf{g}_t^i\|^2.$$

Therefore, we have:

$$\sum_{t=1}^{T} \|\mathbf{e}_t^i\|^2 \leq \frac{2(1-\delta)}{\beta^2 \eta^2 - 2(1-\delta)\eta^4} \sum_{t=0}^{T-2}\|\mathbf{g}_{t+1}^i - \mathbf{g}_t^i\|^2$$

Next, we note the following:

$$\begin{aligned}
\hat{\mathbf{g}}_t^i - \mathbf{g}_t^i &= \Delta_t^i - (\mathbf{g}_t^i - \hat{\mathbf{g}}_{t-1}^i - \eta \mathbf{e}_t^i) + \eta \mathbf{e}_t^i \\
&= \Delta_t^i - \delta_t^i + \eta \mathbf{e}_t^i.
\end{aligned}$$

Therefore, by Young's inequality, we have:

$$\sum_{t=0}^{T-1} \|\hat{\mathbf{g}}_t^i - \mathbf{g}_t^i\|^2 \leq 2(1-\delta)\sum_{t=0}^{T-1}\|\delta_t^i\|^2 + 2\eta^2 \sum_{t=1}^{T-1}\|\mathbf{e}_t^i\|^2$$

$$\leq \frac{8(1-\delta)}{\beta^2 - 2(1-\delta)\eta^2}\sum_{t=0}^{T-2}\|\mathbf{g}_{t+1}^i - \mathbf{g}_t^i\|^2.$$

For the choice of $\eta$ and $\beta$, we choose $\beta = 2\sqrt{1-\delta}\eta$. Since $\beta \equiv 1 - \sqrt{1-\delta}(1+\eta)$, we have:

$$\eta = \frac{\delta}{3\sqrt{1-\delta}(1+\sqrt{1-\delta})}, \quad \beta = \frac{2\delta}{3(1+\sqrt{1-\delta})}.$$

Putting this back, we get the desired results. $\qquad\square$

**Lemma 4.3.** *Given Assumptions 2.1 and 2.3 to 2.5, and let $\eta = \frac{\delta}{3\sqrt{1-\delta}(1+\sqrt{1-\delta})}, \gamma \geq \frac{24\sqrt{2}\ell}{\delta}$, then we have:*

$$\sum_{t=0}^{T-1}\frac{1}{n}\sum_{i=1}^{n}\mathbb{E}\left[\|\mathbf{g}_{t+1}^i - \mathbf{g}_t^i\|^2\right] \leq \frac{80\ell^2}{9\gamma}F_0 + 7T\sigma^2. \tag{16}$$

*Therefore, by Lemma 4.1, we also have:*

$$\sum_{t=0}^{T-1}\frac{1}{n}\sum_{i=1}^{n}\mathbb{E}\left[\|\mathbf{e}_t^i\|^2\right] \leq \frac{5760\ell^2}{\delta^4 \gamma}F_0 + \frac{4536T\sigma^2}{\delta^4}. \tag{17}$$

*Proof.* For simplicity, let's write $r_t^2 = \|\mathbf{x}_{t+1} - \mathbf{x}_t\|^2$. By Assumptions 2.1 and 2.5, we have:

$$\sum_{t=0}^{T-1} \frac{1}{n} \sum_{i=1}^{n} \mathbb{E}\left[\|\mathbf{g}_{t+1}^i - \mathbf{g}_t^i\|^2\right] \le 2\ell^2 \sum_{t=0}^{T-1} \mathbb{E}\left[r_t^2\right] + 4T\sigma^2.$$

Therefore,

$$\sum_{t=0}^{T-1} \frac{1}{n} \sum_{i=1}^{n} \mathbb{E}\left[\|\hat{\mathbf{g}}_t^i - \mathbf{g}_t^i\|^2\right] \le \frac{36(1-\delta)(1+\sqrt{1-\delta})^2}{\delta^2} \sum_{t=0}^{T-2} \frac{1}{n} \sum_{i=1}^{n} \mathbb{E}\left[\|\mathbf{g}_{t+1}^i - \mathbf{g}_t^i\|^2\right]$$

$$\le \frac{72\ell^2(1-\delta)(1+\sqrt{1-\delta})^2}{\delta^2} \sum_{t=0}^{T-2} \mathbb{E}\left[r_t^2\right] + \frac{144T(1-\delta)(1+\sqrt{1-\delta})^2\sigma^2}{\delta^2}$$

$$\le \frac{288\ell^2}{\delta^2} \sum_{t=0}^{T-2} \mathbb{E}\left[r_t^2\right] + \frac{576T\sigma^2}{\delta^2}.$$

By Theorem 3.3, we have:

$$\sum_{t=0}^{T-2} \frac{2\gamma - L}{2} r_t^2 + \sum_{t=0}^{T-2} \langle \hat{\mathbf{g}}_t - \nabla f(\mathbf{x}_t), \mathbf{x}_{t+1} - \mathbf{x}_t \rangle \le F(\mathbf{x}_0) - F(\mathbf{x}_T).$$

Therefore, we have:

$$\sum_{t=0}^{T-1} \frac{\gamma - L}{2} \mathbb{E}\left[r_t^2\right] \le \mathbb{E}\left[F(\mathbf{x}_0) - F(\mathbf{x}_T)\right] + \frac{1}{2\gamma} \sum_{t=0}^{T-1} \mathbb{E}\left[\|\hat{\mathbf{g}}_t - \nabla f(\mathbf{x}_t)\|^2\right]$$

$$\le \mathbb{E}\left[F(\mathbf{x}_0) - F(\mathbf{x}_T)\right] + \frac{1}{\gamma} \sum_{t=0}^{T-1} \mathbb{E}\left[\|\hat{\mathbf{g}}_t - \mathbf{g}_t\|^2\right] + \frac{2T\sigma^2}{\gamma n}$$

$$\le \mathbb{E}\left[F(\mathbf{x}_0) - F(\mathbf{x}_T)\right] + \frac{288\ell^2}{\delta^2\gamma} \sum_{t=0}^{T-2} \mathbb{E}\left[r_t^2\right] + \frac{578T\sigma^2}{\delta^2\gamma}.$$

Now assuming that $\gamma \ge \frac{24\sqrt{2}\ell}{\delta}$, and rearranging, we have:

$$\sum_{t=0}^{T-1} \mathbb{E}\left[r_t^2\right] \le \frac{40}{9\gamma} \mathbb{E}\left[F(\mathbf{x}_0) - F(\mathbf{x}_T)\right] + \frac{1285T\sigma^2}{\delta^2\gamma^2}.$$

Therefore,

$$\sum_{t=0}^{T-1} \frac{1}{n} \sum_{i=1}^{n} \mathbb{E}\left[\|\mathbf{g}_{t+1}^i - \mathbf{g}_t^i\|^2\right] \le \frac{80\ell^2}{9\gamma} \mathbb{E}\left[F(\mathbf{x}_0) - F(\mathbf{x}_T)\right] + \frac{2570\ell^2 T\sigma^2}{\delta^2\gamma^2} + 4T\sigma^2.$$

$\square$

**Theorem 4.4.** *Given Assumptions 2.3 to 2.5, and setting $a_t = 1, \gamma_T = \gamma, \eta = \frac{\delta}{3\sqrt{1-\delta}(1+\sqrt{1-\delta})}$, and taking one initial stochastic gradient step from $\mathbf{x}_0$ to $\mathbf{x}_0'$ if $\psi \not\equiv 0$ and setting*

$$\gamma = \max\left\{ \frac{24\sqrt{2}\ell}{\delta}, \sqrt{\frac{T\sigma^2}{nR_0^2}}, \frac{17T^{1/3}\ell^{1/3}\sigma^{2/3}}{R_0^{2/3}\delta^{4/3}} \right\},$$

*then it takes at most*

$$T = \frac{16R_0^2\sigma^2}{n\varepsilon^2} + \frac{561R_0^2\sqrt{\ell}\sigma}{\delta^2\varepsilon^{3/2}} + \frac{96\sqrt{2}\ell R_0^2}{\delta\varepsilon},$$

*iterations of Algorithm 2 to get $\mathbb{E}\left[F(\bar{\mathbf{x}}_T) - F^\star\right] \le \varepsilon$. Here, $R_0 := \|\mathbf{x}_0 - \mathbf{x}^\star\|$.*

*In particular, this means that with three rounds of uncompressed communication (one for the initial stochastic gradient step, one communicating $\hat{\mathbf{g}}_{-1}$ and one communicating $\bar{\mathbf{g}}$), and $T$ rounds of*

*compressed communications, Algorithm 2 reaches the desired accuracy. Assuming that the cost of sending compressed vectors is* 1 *and sending uncompressed vectors is* $m$, *it costs at most*

$$\frac{16R_0^2\sigma^2}{n\varepsilon^2} + \frac{561R_0^2\sqrt{\ell}\sigma}{\delta^2\varepsilon^{3/2}} + \frac{96\sqrt{2}\ell R_0^2}{\delta\varepsilon} + 3m,$$

*in communications for Algorithm 2 to get:*

$$\mathbb{E}\left[F(\bar{\mathbf{x}}_T) - F^\star\right] \le \varepsilon.$$

*Proof.* For simplicity of notation, let's write $F_t := F(\widetilde{\mathbf{x}}_t) - F(\mathbf{x}^\star)$.

By Lemma 4.3 and Theorem 3.3, we have, when $\gamma \ge \frac{24\sqrt{2}\ell}{\delta}$:

$$\frac{1}{T}\sum_{t=0}^{T-1}\mathbb{E}\left[F_{t+1}\right] \le \frac{\gamma R_0^2}{2T} + \frac{L}{\gamma^2 T}\sum_{t=0}^{T-1}\frac{1}{n}\sum_{i=1}^{n}\mathbb{E}\left[\|\mathbf{e}_t^i\|^2\right] + \frac{\sigma^2}{\gamma n}$$

$$\le \frac{\gamma R_0^2}{2T} + \frac{5760\ell^2 L}{\gamma^3\delta^4 T}F_0 + \frac{4536L\sigma^2}{\gamma^2\delta^4} + \frac{\sigma^2}{\gamma n}.$$

Now we choose $\gamma = \max\left\{\frac{24\sqrt{2}\ell}{\delta}, \frac{11L^{1/4}\ell^{1/2}F_0^{1/4}}{\delta R_0^{1/2}}, \sqrt{\frac{2T\sigma^2}{nR_0^2}}, \frac{21T^{1/3}L^{1/3}\sigma^{2/3}}{R_0^{2/3}\delta^{4/3}}\right\}$, we have:

$$\frac{1}{T}\sum_{t=1}^{T}\mathbb{E}\left[F_t\right] \le \frac{17\ell R_0^2}{\delta T} + \frac{11L^{1/4}\ell^{1/2}F_0^{1/4}R_0^{3/2}}{2\delta T} + \sqrt{\frac{R_0^2\sigma^2}{2nT}} + \frac{21R_0^{4/3}L^{1/3}\sigma^{2/3}}{2T^{2/3}\delta^{4/3}}.$$

Therefore, after:

$$T = \frac{8R_0^2\sigma^2}{2n\varepsilon^2} + \frac{99R_0^2\sqrt{L}\sigma}{\delta^2\varepsilon^{3/2}} + \frac{34\ell R_0^2}{\delta\varepsilon} + \frac{11L^{1/4}\ell^{1/2}F_0^{1/4}R_0^{3/2}}{\delta\varepsilon},$$

iterations of Algorithm 2, we have:

$$\frac{1}{T}\sum_{t=1}^{T}\mathbb{E}\left[F_t\right] \le \varepsilon.$$

Note that this already gives us the desirable convergence rate. We can further simplify the above rates and remove the dependence on $F_0$ by taking one additional stochastic gradient step initially to get $\mathbf{x}_0'$. By Lemma C.3, we have $\mathbb{E}\left[F(\mathbf{x}_0') - F^\star\right] \le LR_0^2 + \frac{R_0\sigma}{\sqrt{2}}$ and $R_0' := \mathbb{E}\left[\|\mathbf{x}_0' - \mathbf{x}^\star\|^2\right] \le 2R_0^2$. Therefore if we start our algorithm at $\mathbf{x}_0'$, then we have:

$$\frac{1}{T}\sum_{t=0}^{T-1}\mathbb{E}\left[F_{t+1}\right] \le \frac{\gamma R_0^2}{T} + \frac{5760\ell^4 R_0^2}{\gamma^3\delta^4 T} + \frac{4073\ell^3 R_0\sigma}{\gamma^3\delta^4 T} + \frac{4536\ell\sigma^2}{\gamma^2\delta^4} + \frac{\sigma^2}{\gamma n}.$$

Note that for the third term, we have the following due Young's inequality and the assumption that $\gamma \ge \frac{24\sqrt{2}\ell}{\delta}$:

$$\frac{4073\ell^3 R_0\sigma}{\gamma^3\delta^4 T} \le \frac{4073\ell^4 R_0^2}{2\gamma^3\delta^4 T} + \frac{4073\ell^2\sigma^2}{2\gamma^3\delta^4 T} \le \frac{4073\ell^4 R_0^2}{2\gamma^3\delta^4 T} + \frac{61\ell\sigma^2}{\gamma^2\delta^3}.$$

Therefore, we have:

$$\frac{1}{T}\sum_{t=0}^{T-1}\mathbb{E}\left[F_{t+1}\right] \le \frac{\gamma R_0^2}{T} + \frac{7797\ell^4 R_0^2}{\gamma^3\delta^4 T} + \frac{4597\ell\sigma^2}{\gamma^2\delta^4} + \frac{\sigma^2}{\gamma n}.$$

Now we pick:

$$\gamma = \max\left\{\frac{24\sqrt{2}\ell}{\delta}, \sqrt{\frac{T\sigma^2}{nR_0^2}}, \frac{17T^{1/3}\ell^{1/3}\sigma^{2/3}}{R_0^{2/3}\delta^{4/3}}\right\},$$

and we have:

$$\frac{1}{T}\sum_{t=1}^{T}\mathbb{E}\left[F_t\right] \le \frac{24\sqrt{2}\ell R_0^2}{\delta T} + \sqrt{\frac{R_0^2\sigma^2}{nT}} + \frac{17R_0^{4/3}\ell^{1/3}\sigma^{2/3}}{T^{2/3}\delta^{4/3}}.$$

Therefore, we need only:

$$T = \frac{16R_0^2\sigma^2}{n\varepsilon^2} + \frac{561R_0^2\sqrt{\ell}\sigma}{\delta^2\varepsilon^{3/2}} + \frac{96\sqrt{2}\ell R_0^2}{\delta\varepsilon}.$$

iterations.

$\square$

## H  EControl WITH VARIABLE STEPSIZE

Consider Theorem 3.3, when the stepsize $\gamma_t$ is changing, we have to upper bound the sum of $\frac{\|e_t\|}{\gamma_{t-1}^2}$. This extra weight has to be handled directly in the analysis.

**Lemma H.1.** *Let* $\eta = \frac{\delta}{3\sqrt{1-\delta}(1+\sqrt{1-\delta})}$*, we have:*

$$\begin{aligned}
\sum_{t=1}^{T}\frac{\|\mathbf{e}_t^i\|^2}{\gamma_{t-1}^2} &\le \frac{81(1-\delta)^2(1+\sqrt{1-\delta})^4}{2\delta^4}\sum_{t=0}^{T-2}\frac{\|\mathbf{g}_{t+1}^i - \mathbf{g}_t^i\|^2}{\gamma_t^2}, \\
\sum_{t=0}^{T-1}\frac{\|\hat{\mathbf{g}}_t^i - \mathbf{g}_t^i\|^2}{\gamma_t^4} &\le \frac{36(1-\delta)(1+\sqrt{1-\delta})^2}{\gamma_0^2\delta^2}\sum_{t=0}^{T-2}\frac{\|\mathbf{g}_{t+1}^i - \mathbf{g}_t^i\|^2}{\gamma_t^2}.
\end{aligned} \tag{22}$$

*Proof.* By the definition of $\mathbf{e}_{t+1}^i$, we have:

$$\mathbf{e}_{t+1}^i := \hat{\mathbf{g}}_t^i - \mathbf{g}_t^i + \mathbf{e}_t^i = \hat{\mathbf{g}}_{t-1}^i + \Delta_t^i - \mathbf{g}_t^i + \mathbf{e}_t^i = \Delta_t^i - \delta_t^i + (1-\eta)\mathbf{e}_t^i,$$

Therefore, by triangular inequality, we have:

$$\|\mathbf{e}_{t+1}^i\| \le (1-\eta)\|\mathbf{e}_t^i\| + \|\Delta_t^i - \delta_t^i\| \le (1-\eta)\|\mathbf{e}_t^i\| + \sqrt{1-\delta}\|\delta_t^i\|,$$

where in the last inequality we used the definition of the compressor. Now divide both sides by $\gamma_t^2$, and noting that $\gamma_t \ge \gamma_{t-1}$, we have:

$$\frac{\|\mathbf{e}_{t+1}^i\|}{\gamma_t^2} \le (1-\eta)\frac{\|\mathbf{e}_t^i\|}{\gamma_{t-1}^2} + \frac{\sqrt{1-\delta}\|\delta_t^i\|}{\gamma_{t-1}^2}.$$

Now by Lemma C.2, we get:

$$\sum_{t=1}^{T}\frac{\|\mathbf{e}_t^i\|^2}{\gamma_{t-1}^2} \le \frac{1-\delta}{\eta^2}\sum_{t=0}^{T-1}\frac{\|\delta_t^i\|^2}{\gamma_{t-1}^2}.$$

Next we note the following:

$$\begin{aligned}
\delta_{t+1}^i &= \mathbf{g}_{t+1}^i - \hat{\mathbf{g}}_t^i - \eta\mathbf{e}_{t+1}^i \\
&= \mathbf{g}_t^i - \hat{\mathbf{g}}_t^i - \eta(\hat{\mathbf{g}}_t^i - \mathbf{g}_t^i + \mathbf{e}_t^i) + \mathbf{g}_{t+1}^i - \mathbf{g}_t^i \\
&= (1+\eta)(\mathbf{g}_t^i - \hat{\mathbf{g}}_t^i) - \eta\mathbf{e}_t^i + \mathbf{g}_{t+1}^i - \mathbf{g}_t^i \\
&= (1+\eta)(\delta_t^i - \Delta_t^i + \eta\mathbf{e}_t^i) - \eta\mathbf{e}_t^i + \mathbf{g}_{t+1}^i - \mathbf{g}_t^i \\
&= (1+\eta)(\delta_t^i - \Delta_t^i) + \eta^2\mathbf{e}_t^i + \mathbf{g}_{t+1}^i - \mathbf{g}_t^i.
\end{aligned}$$

Similar as before, we now apply triangular inequality and definition of the compressor and get:

$$\|\delta_{t+1}^i\| \le (1+\eta)\sqrt{1-\delta}\|\delta_t^i\| + \eta^2\|\mathbf{e}_t^i\| + \|\mathbf{g}_{t+1}^i - \mathbf{g}_t^i\|.$$

Again, we divide both sides by $\gamma_t^2$ and note that $\gamma_t \ge \gamma_{t-1}$:

$$\frac{\|\delta_{t+1}^i\|}{\gamma_t^2} \le (1+\eta)\sqrt{1-\delta}\frac{\|\delta_t^i\|}{\gamma_{t-1}^2} + \frac{\eta^2\|\mathbf{e}_t^i\|}{\gamma_{t-1}^2} + \frac{\|\mathbf{g}_{t+1}^i - \mathbf{g}_t^i\|}{\gamma_t^2}.$$

Let's write $\beta \equiv 1 - (1+\eta)\sqrt{1-\delta}$. Now we apply Lemma C.2 again and Young's inequality, and note that $\delta_0^i = \mathbf{0}$, we get:

$$\sum_{t=0}^{T-1} \frac{\|\delta_t^i\|^2}{\gamma_{t-1}^2} \leq \frac{2}{\beta^2} \sum_{t=0}^{T-2} \left( \frac{\eta^4 \|\mathbf{e}_t^i\|^2}{\gamma_{t-1}^2} + \frac{\|\mathbf{g}_{t+1}^i - \mathbf{g}_t^i\|^2}{\gamma_t^2} \right).$$

Now we plug in the upper bound on the sum of $\|\mathbf{e}_t^i\|$ (and note that $\mathbf{e}_0^i = \mathbf{0}$):

$$\sum_{t=0}^{T-1} \frac{\|\delta_t^i\|^2}{\gamma_{t-1}^2} \leq \frac{2(1-\delta)\eta^2}{\beta^2} \sum_{t=0}^{T-3} \frac{\|\delta_t^i\|^2}{\gamma_{t-1}^2} + \frac{2}{\beta^2} \sum_{t=0}^{T-2} \frac{\|\mathbf{g}_{t+1}^i - \mathbf{g}_t^i\|^2}{\gamma_t^2}.$$

Rearranging, we have:

$$\sum_{t=0}^{T-1} \frac{\|\delta_t^i\|^2}{\gamma_{t-1}^2} \leq \frac{2}{\beta^2 - 2(1-\delta)\eta^2} \sum_{t=0}^{T-2} \frac{\|\mathbf{g}_{t+1}^i - \mathbf{g}_t^i\|^2}{\gamma_t^2}.$$

Therefore, we have:

$$\sum_{t=1}^{T} \frac{\|\mathbf{e}_t^i\|^2}{\gamma_{t-1}^2} \leq \frac{2(1-\delta)}{\beta^2\eta^2 - 2(1-\delta)\eta^4} \sum_{t=0}^{T-2} \frac{\|\mathbf{g}_{t+1}^i - \mathbf{g}_t^i\|^2}{\gamma_t^2}.$$

Next, we note the following:

$$\hat{\mathbf{g}}_t^i - \mathbf{g}_t^i = \Delta_t^i - (\mathbf{g}_t^i - \hat{\mathbf{g}}_{t-1}^i - \eta\mathbf{e}_t^i) + \eta\mathbf{e}_t^i$$
$$= \Delta_t^i - \delta_t^i + \eta\mathbf{e}_t^i.$$

Therefore, by Young's inequality, we have:

$$\sum_{t=0}^{T-1} \frac{\|\hat{\mathbf{g}}_t^i - \mathbf{g}_t^i\|^2}{\gamma_t^2} \leq 2(1-\delta) \sum_{t=0}^{T-1} \frac{\|\delta_t^i\|^2}{\gamma_{t-1}^2} + 2\eta^2 \sum_{t=1}^{T-1} \frac{\|\mathbf{e}_t^i\|^2}{\gamma_{t-1}^2}$$

$$\leq \frac{8(1-\delta)}{\beta^2 - 2(1-\delta)\eta^2} \sum_{t=0}^{T-2} \frac{\|\mathbf{g}_{t+1}^i - \mathbf{g}_t^i\|^2}{\gamma_t^2}.$$

For the choice of $\eta$ and $\beta$, we choose $\beta = 2\sqrt{1-\delta}\eta$. Since $\beta \equiv 1 - \sqrt{1-\delta}(1+\eta)$, we have:

$$\eta = \frac{\delta}{3\sqrt{1-\delta}(1+\sqrt{1-\delta})}, \quad \beta = \frac{2\delta}{3(1+\sqrt{1-\delta})}.$$

Putting this back, we get the desired results. $\qquad\square$

**Lemma H.2.** *Given Assumptions 2.1 and 2.3 to 2.5, and $\eta = \frac{\delta}{3\sqrt{1-\delta}(1+\sqrt{1-\delta})}, \gamma \geq \frac{136\ell}{\delta}$ we have:*

$$\sum_{t=0}^{T-1} \frac{1}{n} \sum_{i=1}^{n} \frac{\mathbb{E}\left[\|\mathbf{g}_{t+1}^i - \mathbf{g}_t^i\|^2\right]}{\gamma_t^2} \leq \frac{32\ell^2 F_0}{\gamma_0^3} + \frac{73988\ell^2}{\gamma_0^2\delta^2} \sum_{t=0}^{T-1} \frac{\sigma^2}{\gamma_t^2}. \tag{23}$$

*Therefore, by Lemma 4.1, we also have:*

$$\sum_{t=1}^{T} \frac{1}{n} \sum_{i=1}^{n} \frac{\mathbb{E}\left[\|\mathbf{e}_t^i\|^2\right]}{\gamma_{t-1}^2} \leq \frac{2^{15}\ell^2 F_0}{\gamma_0^3} + \frac{2^{26}\ell^2}{\gamma_0^2\delta^2} \sum_{t=0}^{T-1} \frac{\sigma^2}{\gamma_t^2}. \tag{24}$$

*Proof.* For simplicity, let's write $r_t^2 = \|\mathbf{x}_{t+1} - \mathbf{x}_t\|^2$. By Assumptions 2.1 and 2.5, we have:

$$\sum_{t=0}^{T-1} \frac{1}{n} \sum_{i=1}^{n} \frac{\mathbb{E}\left[\|\mathbf{g}_{t+1}^i - \mathbf{g}_t^i\|^2\right]}{\gamma_t^2} \leq 2\ell^2 \sum_{t=0}^{T-1} \frac{\mathbb{E}\left[r_t^2\right]}{\gamma_t^2} + \sum_{t=0}^{T-1} \frac{4\sigma^2}{\gamma_t^2}.$$

Therefore,

$$
\sum_{t=0}^{T-1} \frac{1}{n} \sum_{i=1}^{n} \frac{\mathbb{E}\left[\|\hat{\mathbf{g}}_t^i - \mathbf{g}_t^i\|^2\right]}{\gamma_t^4} \leq \frac{36(1-\delta)(1+\sqrt{1-\delta})^2}{\gamma_0^2 \delta^2} \sum_{t=0}^{T-2} \frac{1}{n} \sum_{i=1}^{n} \frac{\mathbb{E}\left[\|\mathbf{g}_{t+1}^i - \mathbf{g}_t^i\|^2\right]}{\gamma_t^2}
$$

$$
\leq \frac{72\ell^2(1-\delta)(1+\sqrt{1-\delta})^2}{\gamma_0^2 \delta^2} \sum_{t=0}^{T-2} \frac{\mathbb{E}\left[r_t^2\right]}{\gamma_t^2} + \frac{144(1-\delta)(1+\sqrt{1-\delta})^2}{\gamma_0^2 \delta^2} \sum_{t=0}^{T-1} \frac{\sigma^2}{\gamma_t^2}
$$

$$
\leq \frac{288\ell^2}{\gamma_0^2 \delta^2} \sum_{t=0}^{T-2} \frac{\mathbb{E}\left[r_t^2\right]}{\gamma_t^2} + \frac{576}{\gamma_0^2 \delta^2} \sum_{t=0}^{T-1} \frac{\sigma^2}{\gamma_t^2}.
$$

Recall the following from the proof of Theorem 3.3 (with $a_t = 1$):

$$
\frac{\gamma_t + \gamma_{t-1} - L}{2} r_t^2 + F(\mathbf{x}_{t+1}) + \langle \hat{\mathbf{g}}_t - \nabla f(\mathbf{x}_t), \mathbf{x}_{t+1} - \mathbf{x}_t \rangle + \frac{\gamma_t - \gamma_{t-1}}{2} \|\mathbf{x}_{t+1} - \mathbf{x}_0\|^2
$$

$$
\leq F(\mathbf{x}_t) + \frac{\gamma_t - \gamma_{t-1}}{2} \|\mathbf{x}_t - \mathbf{x}_0\|^2.
$$

Upper bounding $\langle \hat{\mathbf{g}}_t - \nabla f(\mathbf{x}_t), \mathbf{x}_{t+1} - \mathbf{x}_t \rangle$, and dividing both sides by $\gamma_t^3$ and summing from $t = 0$ to $T - 1$, we have:

$$
\sum_{t=0}^{T-1} \frac{\gamma_t - 2L}{4\gamma_t^3} r_t^2 \leq \sum_{t=0}^{T-1} \frac{F(\mathbf{x}_t) - F(\mathbf{x}_{t+1})}{\gamma_t^3} + 2 \sum_{t=0}^{T-1} \frac{\|\hat{\mathbf{g}}_t - \nabla f(\mathbf{x}_t)\|^2}{\gamma_t^4} + \sum_{t=0}^{T-1} (\beta_t' \rho_t - \beta_t' \rho_{t+1}),
$$

where $\beta_t' \coloneqq \frac{\gamma_t - \gamma_{t-1}}{2\gamma_t^3}$. Note that since $\gamma_t$ is non-decreasing, we also have:

$$
\sum_{t=0}^{T-1} \frac{F(\mathbf{x}_t) - F(\mathbf{x}_{t+1})}{\gamma_t^3} = \frac{F(\mathbf{x}_0) - F(\mathbf{x}^\star)}{\gamma_0^3} - \frac{F(\mathbf{x}_1) - F(\mathbf{x}^\star)}{\gamma_0^3} + \frac{F(\mathbf{x}_1) - F(\mathbf{x}^\star)}{\gamma_1^3} - \frac{F(\mathbf{x}_2) - F(\mathbf{x}^\star)}{\gamma_1^3} + \cdots
$$

$$
+ \frac{F(\mathbf{x}_{T-1}) - F(\mathbf{x}^\star)}{\gamma_{T-1}^3} - \frac{F(\mathbf{x}_T) - F(\mathbf{x}^\star)}{\gamma_{T-1}^3}
$$

$$
\leq \frac{F(\mathbf{x}_0) - F(\mathbf{x}^\star)}{\gamma_0^3} - \frac{F(\mathbf{x}_1) - F(\mathbf{x}^\star)}{\gamma_1^3} + \frac{F(\mathbf{x}_1) - F(\mathbf{x}^\star)}{\gamma_1^3} - \frac{F(\mathbf{x}_2) - F(\mathbf{x}^\star)}{\gamma_2^3} + \cdots
$$

$$
+ \frac{F(\mathbf{x}_{T-1}) - F(\mathbf{x}^\star)}{\gamma_{T-1}^3} - \frac{F(\mathbf{x}_T) - F(\mathbf{x}^\star)}{\gamma_T^3}
$$

$$
\leq \frac{F(\mathbf{x}_0) - F(\mathbf{x}^\star)}{\gamma_0^3}.
$$

Taking expectation on both sides, and applying Assumption 2.1, we have:

$$
\sum_{t=0}^{T-1} \frac{\gamma_t - 2L}{4\gamma_t^3} \mathbb{E}\left[r_t^2\right] \leq \frac{F_0}{\gamma_0^3} + 4 \sum_{t=0}^{T-1} \frac{1}{n} \sum_{i=1}^{n} \frac{\mathbb{E}\left[\|\hat{\mathbf{g}}_t^i - \mathbf{g}_t^i\|^2\right]}{\gamma_t^4} + \sum_{t=0}^{T-1} \frac{8\sigma^2}{\gamma_t^4 n} + \sum_{t=0}^{T-1} (\beta_t' \rho_t - \beta_t' \rho_{t+1}).
$$

Now we use the assumption that $\beta_t'$ is non-increasing and eliminate the last term. Further, we plug in the upper bound for the sum of $\|\hat{\mathbf{g}}_t - \mathbf{g}_t\|^2$, we get:

$$
\sum_{t=0}^{T-1} \frac{\gamma_t - 2L}{4\gamma_t^3} \mathbb{E}\left[r_t^2\right] \leq \frac{F_0}{\gamma_0^3} + \frac{1152\ell^2}{\gamma_0^2 \delta^2} \sum_{t=0}^{T-2} \frac{\mathbb{E}\left[r_t^2\right]}{\gamma_t^2} + \frac{2312}{\gamma_0^2 \delta^2} \sum_{t=0}^{T-1} \frac{\sigma^2}{\gamma_t^2}.
$$

Suppose that $\gamma_t \geq 4L$, then we have:

$$
\sum_{t=0}^{T-1} \frac{1}{\gamma_t^2} \mathbb{E}\left[r_t^2\right] \leq \frac{8F_0}{\gamma_0^3} + \frac{9216\ell^2}{\gamma_0^2 \delta^2} \sum_{t=0}^{T-2} \frac{\mathbb{E}\left[r_t^2\right]}{\gamma_t^2} + \frac{18496}{\gamma_0^2 \delta^2} \sum_{t=0}^{T-1} \frac{\sigma^2}{\gamma_t^2}.
$$

If $\gamma_0 \geq \frac{136\ell}{\delta}$, then we have:

$$
\sum_{t=0}^{T-1} \frac{1}{\gamma_t^2} \mathbb{E}\left[r_t^2\right] \leq \frac{16F_0}{\gamma_0^3} + \frac{36992}{\gamma_0^2 \delta^2} \sum_{t=0}^{T-1} \frac{\sigma^2}{\gamma_t^2}.
$$

Finally, we have:

$$\sum_{t=0}^{T-1} \frac{1}{n} \sum_{i=1}^{n} \frac{\mathbb{E}\left[\|\mathbf{g}_{t+1}^i - \mathbf{g}_t^i\|^2\right]}{\gamma_t^2} \leq \frac{32\ell^2 F_0}{\gamma_0^3} + \frac{73988\ell^2}{\gamma_0^2 \delta^2} \sum_{t=0}^{T-1} \frac{\sigma^2}{\gamma_t^2}.$$

Therefore, by Lemma H.1:

$$\sum_{t=1}^{T} \frac{1}{n} \sum_{i=1}^{n} \frac{\mathbb{E}\left[\|\mathbf{e}_t^i\|^2\right]}{\gamma_{t-1}^2} \leq \frac{2^{15}\ell^2 F_0}{\gamma_0^3} + \frac{2^{26}\ell^2}{\gamma_0^2 \delta^2} \sum_{t=0}^{T-1} \frac{\sigma^2}{\gamma_t^2}.$$

$\square$

**Theorem H.3.** *Given Assumptions 2.3 to 2.5, and we set $a_t = 1$, $\eta := \frac{\delta}{3\sqrt{1-\delta}(1+\sqrt{1-\delta})}$, and we take one initial stochastic gradient step from $\mathbf{x}_0$ to $\mathbf{x}_0'$ if $\psi \not\equiv 0$ and set*

$$\gamma_t = \frac{136\ell}{\delta} + \sqrt{\frac{2t\sigma^2}{nR_0^2}} + \frac{646\ell^{1/3}\sigma^{2/3}t^{1/3}}{R_0^{2/3}\delta^{4/3}},$$

*then it takes at most*

$$T = \frac{288R_0^2\sigma^2}{n\varepsilon^2} + \frac{6692L^{1/2}R_0^2\sigma}{\delta^2\varepsilon^{3/2}} + \frac{552\ell R_0^2}{\delta\varepsilon},$$

*iterations of Algorithm 2 to get $\mathbb{E}\left[F(\bar{\mathbf{x}}_T) - F^\star\right] \leq \varepsilon$.*

*In particular, this means that with three rounds of uncompressed communication (one for the initial stochastic gradient step, one communicating $\hat{\mathbf{g}}_{-1}$ and one communicating $\bar{\mathbf{g}}$), and $T$ rounds of compressed communications, Algorithm 2 reaches the desired accuracy. Assuming that the cost of sending compressed vectors is $1$ and sending uncompressed vectors is $m$, it costs at most*

$$T = \frac{288R_0^2\sigma^2}{n\varepsilon^2} + \frac{6692L^{1/2}R_0^2\sigma}{\delta^2\varepsilon^{3/2}} + \frac{552\ell R_0^2}{\delta\varepsilon} + 3m,$$

*in communications for Algorithm 2 to get:*

$$\mathbb{E}\left[F(\bar{\mathbf{x}}_T) - F^\star\right] \leq \varepsilon.$$

*Proof.* By Lemma H.2 and Theorem 3.3, and setting $\eta = \frac{\delta}{3\sqrt{1-\delta}(1+\sqrt{1-\delta})}$, and assuming that $\gamma_0 \geq \frac{136\ell}{\delta}$ and that $\beta_t'$ is non-increasing in $t$ (this can be easily verified once we give the precise definitions of $\gamma_t$), we have:

$$\frac{1}{T}\sum_{t=0}^{T-1} \mathbb{E}\left[F_{t+1}\right] + \frac{\gamma_{T-1}}{2T}R_T^2 \leq \frac{\gamma_{T-1}}{2T}R_0^2 + \frac{L}{T}\sum_{t=0}^{T-1}\frac{1}{n}\sum_{i=1}^{n}\frac{\mathbb{E}\left[\|\mathbf{e}_t^i\|^2\right]}{\gamma_{t-1}^2} + \sum_{t=0}^{T-1}\frac{\sigma^2}{nT\gamma_{t-1}}$$

$$\leq \frac{\gamma_{T-1}}{2T}R_0^2 + \frac{2^{15}L\ell^2}{\delta^4\gamma_0^3 T}F_0 + \sum_{t=0}^{T-1}\frac{2^{26}L\sigma^2}{\delta^4\gamma_t^2 T} + \sum_{t=0}^{T-1}\frac{\sigma^2}{n\gamma_{t-1}T}.$$

We consider the following stepsize:

$$\gamma_t = \frac{136\ell}{\delta} + \frac{32L^{1/4}\ell^{1/2}F_0^{1/4}}{\delta R_0^{1/2}} + \sqrt{\frac{2t\sigma^2}{nR_0^2}} + \frac{512L^{1/3}\sigma^{2/3}t^{1/3}}{R_0^{2/3}\delta^{4/3}}.$$

First we note that $\gamma_t$ is non-decreasing. Further, it can be verified that with such a choice of $\gamma_t$, we have $\beta_t' = \frac{\gamma_t - \gamma_{t-1}}{2\gamma_t^3}$ is non-increasing in $t$.

Noting that $\sum_{t=0}^{T-1}\frac{1}{t^{1-p}} \leq \frac{1}{p}(T-1)^p$, then we have:

$$\frac{1}{T}\sum_{t=0}^{T-1}\mathbb{E}\left[F_{t+1}\right] \leq \frac{69\ell R_0^2}{\delta T} + \frac{16L^{1/4}\ell^{1/2}F_0^{1/4}}{\delta R_0^{1/2}T} + \frac{384L^{1/3}R_0^{4/3}\sigma^{2/3}}{\delta^{4/3}T^{2/3}} + \frac{3\sqrt{2}R_0\sigma}{\sqrt{nT}}.$$

Therefore, after at most:

$$T = \frac{288R_0^2\sigma^2}{n\varepsilon^2} + \frac{60199L^{1/2}R_0^2\sigma}{\delta^2\varepsilon^{3/2}} + \frac{276\ell R_0^2}{\delta\varepsilon} + \frac{64L^{1/4}\ell^{1/2}F_0^{1/4}}{\delta R_0^{1/2}\varepsilon}.$$

iterations, we have $\mathbb{E}\left[F(\bar{\mathbf{x}}_T)\right] \leq \varepsilon$.

This is already a desirable convergence rate, but we can also eliminate the term dependent on $F_0$, using one initial stochastic gradient step. By Lemma C.3, we have $\mathbb{E}\left[F(\mathbf{x}_0') - F^\star\right] \leq LR_0^2 + \frac{R_0\sigma}{\sqrt{2}}$ and $R_0' := \mathbb{E}\left[\|\mathbf{x}_0' - \mathbf{x}^\star\|^2\right] \leq 2R_0^2$. Therefore if we start our algorithm at $\mathbf{x}_0'$, then we have:

$$\frac{1}{T}\sum_{t=0}^{T-1}\mathbb{E}\left[F_{t+1}\right] \leq \frac{\gamma_{T-1}}{2T}R_0^2 + \frac{2^{15}\ell^4 R_0^2}{\delta^4\gamma_0^3 T} + \frac{2^{15}\ell^3 R_0\sigma}{\delta^4\gamma_0^3 T} + \sum_{t=0}^{T-1}\frac{2^{26}\ell\sigma^2}{\delta^4\gamma_t^2 T} + \sum_{t=0}^{T-1}\frac{\sigma^2}{n\gamma_{t-1}T}.$$

For the third term, due to Young's inequality, we have:

$$\frac{2^{15}\ell^3 R_0\sigma}{\delta^4\gamma_0^3 T} \leq \frac{2^{15}\ell^4 R_0^2}{\delta^4\gamma_0^3 T} + \frac{2^{15}\ell^2\sigma^2}{\delta^4\gamma_0^3 T} \leq \frac{2^{15}\ell^3 R_0\sigma}{\delta^4\gamma_0^3 T} + \frac{241\ell\sigma^2}{\delta^4\gamma_0^2 T}.$$

Therefore, we have:

$$\frac{1}{T}\sum_{t=0}^{T-1}\mathbb{E}\left[F_{t+1}\right] \leq \frac{\gamma_{T-1}}{2T}R_0^2 + \frac{2^{16}\ell^4 R_0^2}{\delta^4\gamma_0^3 T} + \sum_{t=0}^{T-1}\frac{2^{27}\ell\sigma^2}{\delta^4\gamma_t^2 T} + \sum_{t=0}^{T-1}\frac{\sigma^2}{n\gamma_{t-1}T}.$$

Now we pick:

$$\gamma_t = \frac{136\ell}{\delta} + \sqrt{\frac{2t\sigma^2}{nR_0^2} + \frac{646\ell^{1/3}\sigma^{2/3}t^{1/3}}{R_0^{2/3}\delta^{4/3}}}.$$

and after at most:

$$T = \frac{288R_0^2\sigma^2}{n\varepsilon^2} + \frac{6692L^{1/2}R_0^2\sigma}{\delta^2\varepsilon^{3/2}} + \frac{552\ell R_0^2}{\delta\varepsilon}.$$

iterations, we get

$$\mathbb{E}\left[F(\bar{\mathbf{x}}_T) - F^\star\right] \leq \varepsilon$$

.

$\square$

## I   ANALYSIS OF THE REAL ITERATES

In this section, we present an analysis of the real iterates generated by Algorithm 1, which can be immediately combined with our analysis in Section 4 and give the convergence guarantee for Algorithm 2 purely in terms of the real iterates $\mathbf{x}_t$. We note that this analysis does not rely on the virtual iterates $\widetilde{\mathbf{x}}_t$ at all, and is therefore also applicable to the basic proximal algorithm without dual averaging. We believe that this analysis might be of independent interest.

We first note that the guarantees for the real iterates is weaker than that of Theorem 3.3.

**Theorem I.1.** *Given Assumptions 3.1, 2.3 and 2.4, then for any $\mathbf{x} \in \mathrm{dom}\psi$, we have:*

$$\sum_{t=0}^{T-1}\mathbb{E}\left[a_t(F(\mathbf{x}_{t+1}) - F(\mathbf{x})) + \frac{\gamma_{t-1} - 2a_t L}{4}\|\mathbf{x}_{t+1} - \mathbf{x}_t\|^2\right] \leq \frac{\gamma_{T-1}}{2}\|\mathbf{x} - \mathbf{x}_0\|^2 + 2\sum_{t=1}^{T}\frac{\mathbb{E}\left[\|\mathbf{e}_t\|^2\right]}{\gamma_{t-1}} + 2\sum_{t=0}^{T-1}\frac{a_t^2\sigma_\mathbf{g}^2}{\gamma_{t-1}}. \tag{25}$$

*Proof.* By the definition of $\Phi_t$, we have for any $\mathbf{x} \in \mathrm{dom}(\psi)$:

$$\Phi_t(\mathbf{x}) \geq \Phi_t^\star + \frac{1}{2}\|\mathbf{x} - \mathbf{x}_{t+1}\|^2.$$

We also have:

$$
\begin{aligned}
\Phi_t(\mathbf{x}) &= \sum_{k=0}^{t} a_k(f(\mathbf{x}_k) + \langle \hat{\mathbf{g}}_k, \mathbf{x} - \mathbf{x}_s \rangle + \psi(\mathbf{x})) + \frac{\gamma_t}{2}\|\mathbf{x} - \mathbf{x}_0\|^2 \\
&= \sum_{k=0}^{t} a_k(f(\mathbf{x}_k) + \langle \nabla f(\mathbf{x}_k), \mathbf{x} - \mathbf{x}_k \rangle + \psi(\mathbf{x})) + \sum_{k=0}^{t} a_k \langle \hat{\mathbf{g}}_k - \nabla f(\mathbf{x}_k), \mathbf{x} - \mathbf{x}_k \rangle + \frac{\gamma_t}{2}\|\mathbf{x} - \mathbf{x}_0\|^2 \\
&\overset{(i)}{\leq} \sum_{k=0}^{t} a_k F(\mathbf{x}) + \sum_{k=0}^{t} a_k \langle \hat{\mathbf{g}}_k - \nabla f(\mathbf{x}_k), \mathbf{x} - \mathbf{x}_k \rangle + \frac{\gamma_t}{2}\|\mathbf{x} - \mathbf{x}_0\|^2,
\end{aligned}
$$

where in $(i)$ we used the convexity of $f$. Taking expectations on both sides, we get:

$$
\mathbb{E}\left[\Phi_t(\mathbf{x})\right] \leq \sum_{k=0}^{t} a_k F(\mathbf{x}) + \frac{\gamma_t}{2}\|\mathbf{x} - \mathbf{x}_0\|^2.
$$

Now by the definition of $\mathbf{x}_{t+1}$:

$$
\begin{aligned}
\Phi_t^{\star} &= \Phi_{t-1}(\mathbf{x}_{t+1}) + a_t(f(\mathbf{x}_t) + \langle \hat{\mathbf{g}}_t, \mathbf{x}_{t+1} - \mathbf{x}_t \rangle + \psi(\mathbf{x}_{t+1})) + \frac{\gamma_t - \gamma_{t-1}}{2}\|\mathbf{x}_{t+1} - \mathbf{x}_0\|^2 \\
&\overset{(ii)}{\geq} \Phi_{t-1}^{\star} + \frac{\gamma_{t-1}}{2}\|\mathbf{x}_{t+1} - \mathbf{x}_t\|^2 + a_t(f(\mathbf{x}_t) + \langle \hat{\mathbf{g}}_t, \mathbf{x}_{t+1} - \mathbf{x}_t \rangle + \psi(\mathbf{x}_{t+1})) \\
&= \Phi_{t-1}^{\star} + \frac{\gamma_{t-1}}{2}\|\mathbf{x}_{t+1} - \mathbf{x}_t\|^2 + a_t(f(\mathbf{x}_t) + \langle \nabla f(\mathbf{x}_t), \mathbf{x}_{t+1} - \mathbf{x}_t \rangle + \psi(\mathbf{x}_{t+1})) + a_t\langle \hat{\mathbf{g}}_t - \nabla f(\mathbf{x}_t), \mathbf{x}_{t+1} - \mathbf{x}_t \rangle \\
&\overset{(iii)}{\geq} \Phi_{t-1}^{\star} + \frac{\gamma_{t-1} - a_t L}{2}\|\mathbf{x}_{t+1} - \mathbf{x}_t\|^2 + a_t F(\mathbf{x}_{t+1}) + a_t\langle \hat{\mathbf{g}}_t - \nabla f(\mathbf{x}_t), \mathbf{x}_{t+1} - \mathbf{x}_t \rangle,
\end{aligned}
$$

where in $(ii)$ we used the 1-strong convexity of $\Phi_t$ and in $(iii)$ we used Assumption 2.4.

Now rearranging and summing from $t = 0$ to $T - 1$, we get:

$$
\begin{aligned}
&\sum_{t=0}^{T-1} \mathbb{E}\left[a_t F(\mathbf{x}_{t+1}) + \frac{\gamma_{t-1} - a_t L}{2}\|\mathbf{x}_{t+1} - \mathbf{x}_t\|^2\right] \\
&\leq \mathbb{E}\left[\Phi_{T-1}^{\star}\right] - \sum_{t=0}^{T-1} \mathbb{E}\left[a_t\langle \hat{\mathbf{g}}_t - \nabla f(\mathbf{x}_t), \mathbf{x}_{t+1} - \mathbf{x}_t \rangle\right] \\
&\leq \mathbb{E}\left[\Phi_{T-1}^{\star}\right] - \frac{\gamma_{T-1}}{2}\mathbb{E}\left[\|\mathbf{x} - \mathbf{x}_T\|^2\right] - \sum_{t=0}^{T-1} a_t\mathbb{E}\left[\langle \hat{\mathbf{g}}_t - \nabla f(\mathbf{x}_t), \mathbf{x}_{t+1} - \mathbf{x}_t \rangle\right] \\
&\leq \sum_{t=0}^{T-1} a_t F(\mathbf{x}) + \sum_{t=0}^{T-1} a_t\mathbb{E}\left[\langle \hat{\mathbf{g}}_t - \nabla f(\mathbf{x}_t), \mathbf{x} - \mathbf{x}_t \rangle\right] + \frac{\gamma_{T-1}}{2}\mathbb{E}\left[\|\mathbf{x} - \mathbf{x}_0\|^2\right] \\
&\quad - \frac{\gamma_{T-1}}{2}\mathbb{E}\left[\|\mathbf{x} - \mathbf{x}_T\|^2\right] - \sum_{t=0}^{T-1} a_t\mathbb{E}\left[\langle \hat{\mathbf{g}}_t - \nabla f(\mathbf{x}_t), \mathbf{x}_{t+1} - \mathbf{x}_t \rangle\right].
\end{aligned}
$$

Rearranging, we get:

$$
\begin{aligned}
&\sum_{t=0}^{T-1} \mathbb{E}\left[a_t(F(\mathbf{x}_{t+1}) - F(\mathbf{x})) + \frac{\gamma_{t-1} - a_t L}{2}\|\mathbf{x}_{t+1} - \mathbf{x}_t\|^2\right] \\
&\leq \frac{1}{2}(\|\mathbf{x} - \mathbf{x}_0\|^2 - \mathbb{E}\left[\|\mathbf{x} - \mathbf{x}_T\|^2\right]) + \sum_{t=0}^{T-1} a_t\mathbb{E}\left[\langle \hat{\mathbf{g}}_t - \nabla f(\mathbf{x}_t), \mathbf{x} - \mathbf{x}_{t+1} \rangle\right].
\end{aligned}
$$

Note that by the definition of $\mathbf{e}_t$, we have:

$$
\begin{aligned}
\sum_{t=0}^{T-1} a_t \langle \hat{\mathbf{g}}_t - \nabla f(\mathbf{x}_t), \mathbf{x} - \mathbf{x}_{t+1} \rangle &= \sum_{t=0}^{T-1} \langle \mathbf{e}_{t+1} - \mathbf{e}_t, \mathbf{x} - \mathbf{x}_{t+1} \rangle \\
&= \sum_{t=0}^{T-1} \langle \mathbf{e}_{t+1}, \mathbf{x} - \mathbf{x}_{t+1} \rangle - \langle \mathbf{e}_t, \mathbf{x} - \mathbf{x}_t \rangle + \langle \mathbf{e}_t, \mathbf{x}_{t+1} - \mathbf{x}_t \rangle \\
&= \langle \mathbf{e}_T, \mathbf{x} - \mathbf{x}_T \rangle + \sum_{t=0}^{T-1} \langle \mathbf{e}_t, \mathbf{x}_{t+1} - \mathbf{x}_t \rangle \\
&\leq \frac{\|\mathbf{e}_T\|^2}{2\gamma_{T-1}} + \frac{\gamma_{T-1}\|\mathbf{x} - \mathbf{x}_T\|^2}{2} + \sum_{t=1}^{T-1} \left( \frac{2\|\mathbf{e}_t\|^2}{\gamma_{t-1}} + \frac{\gamma_{t-1}\|\mathbf{x}_t - \mathbf{x}_{t+1}\|^2}{8} \right).
\end{aligned}
$$

Taking expectation on both sides, and noting that the noise $\mathbf{g}_t - \nabla f(\mathbf{x}_t)$ is independent on both $\mathbf{x}$ and $\mathbf{x}_t$, we have:

$$
\begin{aligned}
&\sum_{t=0}^{T-1} a_t \mathbb{E}\left[ \langle \hat{\mathbf{g}}_t - \nabla f(\mathbf{x}_t), \mathbf{x} - \mathbf{x}_{t+1} \rangle \right] \\
&\leq \frac{\|\mathbf{e}_T\|^2}{2\gamma_{T-1}} + \frac{\gamma_{T-1}\|\mathbf{x} - \mathbf{x}_T\|^2}{2} + \sum_{t=1}^{T-1} \frac{2\|\mathbf{e}_t\|^2}{\gamma_{t-1}} + \sum_{t=1}^{T-1} \frac{\gamma_{t-1}\|\mathbf{x}_t - \mathbf{x}_{t+1}\|^2}{4} + 2\sum_{t=0}^{T-1} \frac{a_t^2 \sigma_{\mathbf{g}}^2}{\gamma_{t-1}}.
\end{aligned}
$$

Now we put these together and get:

$$
\sum_{t=0}^{T-1} \mathbb{E}\left[ a_t(F(\mathbf{x}_{t+1}) - F(\mathbf{x})) + \frac{\gamma_{t-1} - 2a_t L}{4} \|\mathbf{x}_{t+1} - \mathbf{x}_t\|^2 \right] \leq \frac{\gamma_{T-1}}{2} \|\mathbf{x} - \mathbf{x}_0\|^2 + 2\sum_{t=1}^{T} \frac{\mathbb{E}\left[\|\mathbf{e}_t\|^2\right]}{\gamma_{t-1}} + 2\sum_{t=0}^{T-1} \frac{a_t^2 \sigma_{\mathbf{g}}^2}{\gamma_{t-1}}.
$$

$\square$

*Remark* I.2. Comparing to Theorem 3.3, we note that the key difference here is that the error in Equation (25) is $2\sum_{t=1}^{T} \frac{\mathbb{E}[\|\mathbf{e}_t\|^2]}{\gamma_{t-1}}$, while in Equation (12) it is $L\sum_{t=1}^{T} \frac{\mathbb{E}[\|\mathbf{e}_t\|^2]}{\gamma_{t-1}^2}$. The $\frac{L}{\gamma_{t-1}}$ multiplicative difference here is crucial and allows the stepsize $\gamma_t$ to control the errors much more effectively. Therefore, Theorem I.1 would lead to a weaker convergence guarantee.

With this, we can now directly combine Theorem I.1 with Lemma 4.3 to obtain the following convergence guarantee for Algorithm 2 in terms of the real iterates $\mathbf{x}_t$. For simplicity, we use the fixed stepsizes $\gamma_t = \gamma$.

**Theorem I.3.** *Given Assumptions 2.3 to 2.5, and we set $a_t = 1, \eta := \frac{\delta}{3\sqrt{1-\delta}(1+\sqrt{1-\delta})}$, and we set:*

$$
\gamma = \max\left\{ \frac{24\sqrt{2}\ell}{\delta}, \frac{32\ell^{2/3}F_0^{1/3}}{\delta^{4/3}R_0^{2/3}}, \frac{135\sigma\sqrt{T}}{\delta^2 R_0} \right\},
$$

*then it takes at most:*

$$
T = \frac{72900 R_0^2 \sigma^2}{\delta^4 \varepsilon^2} + \frac{48\sqrt{2}\ell R_0^2}{\delta \varepsilon} + \frac{64(\ell R_0^2)^{2/3}F_0^{1/3}}{\delta^{4/3}\varepsilon}, \tag{26}
$$

*iterations of Algorithm 2 to get $\frac{1}{T}\sum_{t=0}^{T-1}(F(\mathbf{x}_{t+1}) - F^\star) \leq \varepsilon$.*

*Proof.* We plug Equation (17) into Equation (25), and assume that $\gamma \geq \frac{24\sqrt{2}\ell}{\delta}$, and get:

$$
\frac{1}{T}\sum_{t=0}^{T-1}(F(\mathbf{x}_{t+1}) - F^\star) \leq \frac{\gamma R_0^2}{2T} + \frac{11520\ell^2}{\delta^4\gamma^2 T}F_0 + \frac{9074\sigma^2}{\delta^4\gamma}.
$$

Now we set

$$
\gamma = \max\left\{ \frac{24\sqrt{2}\ell}{\delta}, \frac{32\ell^{2/3}F_0^{1/3}}{\delta^{4/3}R_0^{2/3}}, \frac{135\sigma\sqrt{T}}{\delta^2 R_0} \right\},
$$

and we have:

$$\frac{1}{T}\sum_{t=0}^{T-1}(F(\mathbf{x}_{t+1}) - F^\star) \leq \frac{24\sqrt{2}\ell R_0^2}{\delta T} + \frac{32(\ell R_0^2)^{2/3}F_0^{1/3}}{\delta^{4/3}T} + \frac{135 R_0\sigma}{\delta^2\sqrt{T}}.$$

Therefore, it takes at most:

$$T = \frac{72900 R_0^2\sigma^2}{\delta^4\varepsilon^2} + \frac{48\sqrt{2}\ell R_0^2}{\delta\varepsilon} + \frac{64(\ell R_0^2)^{2/3}F_0^{1/3}}{\delta^{4/3}\varepsilon},$$

iterations of Algorithm 2 to get:

$$\frac{1}{T}\sum_{t=0}^{T-1}(F(\mathbf{x}_{t+1}) - F^\star) \leq \varepsilon.$$

$\square$

*Remark* I.4. We emphasize that here we only achieved an $\mathcal{O}(\frac{1}{\delta^{4/3}\varepsilon})$ convergence rate in the deterministic term, which is worse than the $\mathcal{O}(\frac{1}{\delta\varepsilon})$ rate achieved in Theorem 4.4 in terms of $\delta$. Perhaps more importantly, in the stochastic case ($\sigma^2 > 0$), we only achieve a $\mathcal{O}(\frac{1}{\delta^4\varepsilon^2})$ rate, which does not improve linearly as $n$ increases and is not delta-free, unlike the rate in Theorem 4.4 and Theorem H.3. It is unclear whether this limitation is a fundamental property of the algorithm or an artifact of the analysis. We leave it for future work to resolve this question.

*Remark* I.5. We also briefly note that the rate in Theorem I.3 can be slightly improved using the restart strategy and a more careful analysis of the number of steps and parameter settings in each stage. This way we can remove the $\mathcal{O}(\frac{1}{\delta^{4/3}\varepsilon})$ term, and instead get a $\mathcal{O}(\frac{1}{\delta^{4/3}\varepsilon^{2/3}})$ term overall. We will however have to assume that $\mathrm{dom}\psi$ is bounded, and do $\mathcal{O}(\log\frac{1}{\varepsilon})$ number of restarts which requires one full communication at each stage. For simplicity, we omit the details here.

