# OpenReview forum: "Composite Optimization with Error Feedback: the Dual Averaging Approach"
_ICLR.cc/2026/Conference — ICLR 2026 Poster_

### Official Review · Reviewer_Hh9y · 2025-10-15

**Soundness:** 2
**Presentation:** 3
**Contribution:** 2
**Rating:** 4
**Confidence:** 3

**Summary:**

This paper addresses the failure of standard EF methods in composite optimization, where the objective includes a non-smooth component. The authors argue that the proximal step used in this setting disrupts the additive structure essential for traditional EF analysis. To resolve this, they propose a novel algorithm that combines Dual Averaging with a modern EF variant (EControl). This approach successfully restores the necessary structure, enabling the first strong convergence guarantees for composite optimization with communication compression.

**Strengths:**

- Fills a gap in optimization theory by providing the theoretically-backed EF method for the general composite setting.

- The use of Dual Averaging is a conceptually clean and logical solution that directly targets the structural issue at the core of the problem.

**Weaknesses:**

- The claims of the paper are not backed by strong experimental evidence. The experiments mentioned are not detailed enough to validate the practical advantages of the proposed method over existing techniques.

- While the authors suggest their analysis provides a versatile template for other domains (e.g., safe reinforcement learning), the paper does not demonstrate these applications.

**Questions:**

N/A

---

> ### Author Response · Authors · 2025-11-25
>
> We thank the reviewer for the review and we address the concerns in the following:
>
> - We wish to point out that our current paper is mostly theory-focused, and we addressed an important gap in the theory of the error feedback and virtual iteration using our novel inexact dual averaging framework. The experiments are intended to verify our theoretical findings. It would indeed be an interesting future work to comprehensively and rigorously compare our methods to existing methods experimentally, but we believe that this is beyond the scope of the current paper.
>
> - We note that the composite optimization setting naturally covers many domains of applications. For its connection to safe reinforcement learning, we refer to [1].
>
> [1] Islamov, Rustem, Yarden As, and Ilyas Fatkhullin. "Safe-EF: Error Feedback for Non-smooth Constrained Optimization." Forty-second International Conference on Machine Learning.

---

### Official Review · Reviewer_nFER · 2025-10-19

**Soundness:** 3
**Presentation:** 3
**Contribution:** 3
**Rating:** 6
**Confidence:** 4

**Summary:**

The work deals with reducing the communication in distributed optimization cost via compression, in the setting where there is an additional regularization term whose proximity operator is applied by the server.

**Strengths:**

Reducing the communication cost of distributed optimization is an important topic. Combining compression and stochastic gradients is not fully understood and this work is a step forward in this direction.

**Weaknesses:**

** I have the following concerns

1)  "In the convex regime, it is believed that the convergence of proximal EF21 critically relies on the bounded domain assumption, which we do not assume in our work" This is not correct. There is no such restriction for EF21. The Islamov et al. paper is for non-smooth functions, which is a different setting.

2) "EF21, which is more closely related to gradient-difference compression methods". I don't know why you claim that EF21 might not be really an EF method. The control variates are defined recursively, so they accumulates past compression errors.

3) A refined analysis of EF21 in the composite case is in Condat et al. “EF-BV: A Unified Theory of Error Feedback and Variance Reduction Mechanisms for Biased and Unbiased Compression in Distributed Optimization,” Neurips 2022

4) It seems that the focus of the work is about dealing with stochastic gradient errors and linear speedup with respect to such errors (linear speedup with respect to compression errors with independent unbiased compressors is equally important but is not addressed).

5) Explain why convexity is critical to the analysis. Convexity is a restrictive assumption. This is fine to me, but on the other hand in convex settings, Nesterov acceleration becomes available. So, there is an important gap between your guarantees and the convergence speed that can be obtained if $\sigma\rightarrow 0$ and $\delta\rightarrow 1$.

** Typos and minor comments

* Beznosikov et al: the reference is given twice.

* Fatkhullin et al. 2021: published in Journal of Machine Learning Research, 26, 2025

* Algorithm 1 line 4: :=:=

* unlesss

* There should be a period at the end of equations, e.g. 15, 16, 17

**Questions:**

1) The $f_i$ can be non-convex? It seems that yes, but it is important to state it clearly.

2) What is the real challenge in dealing with $\Psi$? In algorithms such as EF21 or DIANA, dealing with $\Psi$ is almost trivial, using nonexpansiveness of the prox. (Exploiting strong convexity of Psi to get linear convergence is less straightforward). This should be clearly explained as this is the main contribution of the work.

3) Can you prove linear convergence if $f$ or $\Psi$ is strongly convex?

---

> ### Author Response · Authors · 2025-11-25
>
> We thank the reviewer for the positive feedback on our paper and address the concerns in the following:
>
> - We wish to point out that Islamov et al. gave the convex case analysis of EF21 in the smooth setting in Appendix C, even though the paper mostly focused on the non-smooth setting (and demonstrated the non-convergence of EF21 in the non-smooth setting). We are not aware of other works that consider proximal EF21 in the smooth and general convex setting that we consider.
>
> - We thank the reviewer for pointing out the EF-BV paper, and we have updated the related work section in Appendix A to discuss it. Indeed, Condat et al discussed the analysis of EF21 in the composite setting, under the PL condition and KL condition with non-zero constants. Under these conditions, the convergence of proximal EF21 does not rely on the bounded domain assumption. It is, however, unclear to us whether these analyses can be extended to the general convex case as is.
>
> - Regarding the convexity assumption: indeed, $f_i$ does not have to be individually convex, and we have updated the paper to clearly state it. It is true that Nesterov acceleration might be possible under the convexity assumption. Handling the increasing stepsizes in the typical accelerated methods along with the EControl mechanism seems challenging and might require some special consideration regarding the control parameter $\eta$ and it would be interesting future work to see if our method can be combined with Nesterov acceleration. Regarding strong convexity and linear convergence, it would require a modification of Theorem 3.3 and its analysis to account for the strong convexity of $F$. We left this out in the current paper due to time and space constraints, and it would be interesting to include them in future works.
>
> - Regarding the difficulty in handling the composite part: we have detailed the challenges of handling the composite setting with the virtual iteration framework in Section 1.1. We pointed out that the virtual iteration framework is essentially the only tool that has been applied to study the classical EF framework a la Seide et al, and that in its basic form, it is incompatible with the composite setting. In Section 1.2 we outlined why Dual Averaging should be a natural remedy for it, which led to our proposed algorithm and analysis. We realize that handling the composite part for methods like EF21 is less tricky compared to their basic uncomposite versions, however, as we have discussed, it seems that are still some technical difficulties for EF21 with the general convex setting, even when using full gradients. Nevertheless, we are mostly interested in extending the classical EF framework a la Seide et al and the virtual iteration framework a la Mania et al, for which we proposed the inexact dual averaging framework.

---

> > ### Comment · Reviewer_nFER · 2025-11-27
> >
> > I have read the different reviews and rebuttals. Thank you for taking my comments into account. I keep my score unchanged.

---

### Official Review · Reviewer_hvZz · 2025-10-29

**Soundness:** 3
**Presentation:** 3
**Contribution:** 2
**Rating:** 4
**Confidence:** 4

**Summary:**

This paper studies distributed **composite** optimization, $\min_x f(x)+\psi(x)$ with smooth $f$ and (possibly) non-smooth/prox-friendly $\psi$, under **compression with error feedback (EF)**. The authors combine **Dual Averaging (DA)** with **EControl** and analyze “virtual iterates,” claiming *“for the first time a strong convergence analysis for composite optimization with error feedback”* (Abstract; Sec. 1). They give an inexact-DA analysis template and experiments on synthetic $\ell_1$-regularized softmax and FashionMNIST with Top-$k$ compression, comparing against proximal EF and proximal EF21.

**Strengths:**

* **Clarity of Presentation:** The paper is well-written, logically structured, and easy to follow. The theoretical development is presented step-by-step, making the complex analysis accessible. The distinction between the real and virtual iterates, a key and often confusing point in EF literature, is handled with consistent clarity.
- **Conceptual:** Clear identification that naïve EF analyses break in the composite case; DA is a natural vehicle to handle the prox term. (Sec. 3–4 table of contents suggests an organized build-up.)
- **Analysis template:** An inexact-DA framework and a sampling scheme for virtual iterates (Secs. 3, 3.1) that could be reusable beyond EControl.
- **Mechanistic clarity:** A separate technical section on EControl’s properties (App. G, H), and a discussion of real vs virtual iterates (App. I).
- **Experiments** provide both synthetic composite and a FashionMNIST $\ell_1$-regularized logistic example with Top-$k$ compression (δ=0.1), and include a *real vs virtual iterate* comparison. (Sec. 5; Fig. 1–2).

**Weaknesses:**

- **Overstated novelty (major):** The paper claims the first “strong convergence analysis for composite optimization with EF.” However, **composite EF** existed well before:
  - EC-ProxSGD and **EC-RDA (dual averaging)** handle composite finite-sum with error compensation. (OPT 2020).
  - **EC-LSVRG/Quartz/SDCA** support composite settings with linear/convex guarantees (arXiv 2021).
  - **EF-BV** (NeurIPS 2022) treats composite objectives with an explicit proximal step within a unified EF/variance-reduction theory.
  - **Eco-FedSplit** (ICASSP 2022) integrates error-compensated compression with proximal FL splitting.
  - **EF21-Prox** makes EF21 applicable to composite problems and states convergence (JMLR/ArXiv 2021).
  The manuscript must **qualify** its “first” claim (e.g., *first DA+EControl-based virtual-iterate rate under assumptions X, Y*), and clearly delineate what “strong” means vs prior composite-EF rates.

[1] Qian, Xun, et al. "Error compensated proximal SGD and RDA." _Proc. 12th Annu. Workshop Optim. Mach. Learn._. 2020.

[2] Qian, Xun, et al. "Error compensated loopless SVRG for distributed optimization." _Workshop on Optimization for Machine Learning_. 2020.

[3] Qian, Xun, et al. "Error compensated loopless svrg, quartz, and sdca for distributed optimization." _arXiv preprint arXiv:2109.10049_ (2021).

[4] Condat, Laurent, Kai Yi, and Peter Richtárik. "EF-BV: A unified theory of error feedback and variance reduction mechanisms for biased and unbiased compression in distributed optimization." _Advances in Neural Information Processing Systems_ 35 (2022): 17501-17514.

[5] Khirirat, Sarit, Sindri Magnússon, and Mikael Johansson. "Eco-fedsplit: Federated learning with error-compensated compression." _ICASSP 2022-2022 IEEE International Conference on Acoustics, Speech and Signal Processing (ICASSP)_. IEEE, 2022.

[6] Fatkhullin, Ilyas, et al. "EF21 with bells & whistles: Practical algorithmic extensions of modern error feedback." _arXiv preprint arXiv:2110.03294_ (2021).


- **Related-work coverage (major):** Sec. A emphasizes uncomposite EF and Safe-EF, but does **not** cite EC-ProxSGD/EC-RDA (OPT’20), EC-LSVRG/Quartz/SDCA (’21), EF-BV (NeurIPS’22), Eco-FedSplit (ICASSP’22). Add these and discuss differences in assumptions (e.g., convexity, finite-sum vs online), or the novelty claim is misleading.
- **Baselines (major):** Experiments compare to “proximal EF” and “proximal EF21,” but **omit** EC-ProxSGD/EC-RDA, EF-BV, and EC-LSVRG/Quartz/SDCA—precisely the most relevant **composite+EF** competitors. Include these, or justify why not.
- **Virtual vs real iterate gap:** Theory primarily addresses a sample of **virtual** iterates; experiments show similarity to **real** iterates on one setup. Provide bounds bridging real iterates (App. I) to the main claims, or broaden empirical validation.
- **Scope/assumptions clarity:** Precisely state smoothness/convexity/stochastic-noise assumptions used to obtain the claimed “strong” rate, and contrast to prior composite-EF (e.g., whether rates are stationarity vs suboptimality, and dependence on $\delta$, $n$, variance). The Abstract currently suggests generality that theorems may not cover.
* **Gap Between Virtual and Real Iterate Guarantees:** The strong theoretical results, including linear speedup, are proven only for the *virtual iterates*. The analysis for the *real iterates* in Appendix I is substantially weaker, yielding a rate of $\mathcal{O}(1/(\delta^4 \epsilon^2))$ that lacks linear speedup and has a much worse dependency on the compression parameter $\delta$. The experiments show that real and virtual iterates perform similarly, which suggests the analysis for the real iterates is not tight. The proposed sampling procedure to recover the virtual iterate guarantee is impractical as it requires storing uncompressed cumulative gradients and an extra round of uncompressed communication, undermining the method's efficiency goals.

**Questions:**

1. Please **restate** the novelty claim precisely: relative to EC-RDA (dual averaging) and EC-ProxSGD (OPT’20), what is mathematically new in your DA+EControl analysis (assumptions, iterates, rate form)? How does “strong convergence” differ from prior composite-EF rates? Could the authors please clarify their novelty claim in light of these specific prior works? Why were these methods not cited or compared against?
2. Can you add **EC-ProxSGD/EC-RDA, EF-BV, and EC-LSVRG/Quartz/SDCA** as baselines—or explain incompatibilities?
3. Is there a **theoretical link** from virtual to **real** iterates beyond empirical similarity (Fig. 1b)? Can you deliver a bound for the real iterate under your stepsize choices?
4.  The analysis of real iterates in Appendix I is much weaker than the analysis of virtual iterates, yet experiments suggest their performance is nearly identical. What is the fundamental analytical barrier preventing a tighter analysis for the real iterates? Is there a specific step in the proof (e.g., handling the proximal operator directly) where the analysis loses tightness?
5.  Given that Qian et al. (2020) already analyzed an error-compensated RDA, could you elaborate on the specific contributions of your inexact dual averaging analysis template (Section 3)? Does it address limitations in the analysis of Qian et al. or offer a different perspective that enables the stronger (virtual iterate) result?

**Details Of Ethics Concerns:**

No ethics concerns identified.

---

> ### Author Response · Authors · 2025-11-25
>
> We thank the reviewer for the review, and we address the reviewer's concerns in the following:
>
> - Regarding the papers that the reviewer pointed out: we thank the reviewer for pointing out these references, and we have updated our related work section in Appendix A to discuss the relevant literature. In short, we noted that EC-ProxSGD and EC-RDA consider the finite-sum stochastic setting, which differs from ours. For both these methods, the known upper bounds differ from the upper bounds known for the uncomposite counterparts in their dependence on $\delta$, and EC-RDA requires, in addition to smoothness, a series of bounded gradient assumptions which we do not assume. EC-LSVRG, EC-Quartz, and EC-SDCA are variance reduction variants of EF, and they consider different settings than ours as well, where EC-LSVRG requires periodic access to full gradients, while EC-Quartz and EC-SDCA are primal dual methods designed for a specific form of objectives and assume access to first-order information of certain conjugates of the regularizer. EF-BV indeed provided a composite analysis of EF21, but it's restricted to the full gradient regime and assumes either PL or KL conditions. It is unclear to us whether the analysis presented therein can be extended to the general convex setting considered in this work. We note that we did not include Eco-FedSplit in the update, because it is an EF-enhanced variant of FedProx, where they do not consider composite optimization in our sense. We have also stated clearly that we achieve, for the first time, a convergence analysis for convex composite optimization with EF that matches the best-known results in the uncomposite setting, thereby clarifying what we mean by the first strong convergence analysis.
>
> - Experimentally, we compare our algorithm to proximal EF, proximal EF21, and proximal EControl. In our setting, EC-ProxSGD should be instantiated with $p=0$ since we do not transmit full messages periodically and in such a case, it's identical to proximal EF except for being shifted by the initial gradient. EF-BV in our setting should be instantiated with $\lambda=\upsilon=1$ and is therefore reduced to proximal EF21. We do not compare to EC-LSVRG, EC-Quartz and EC-SCDA because they consider different problems than ours as explained before. We believe that a much more comprehensive and rigurous experimental test suite would be very interesting future work.
>
> - It is unclear for the moment if and how the analysis of the real iterate can be improved. Technically, when performing the virtual iteration analysis for the real iterates, one has to upper bound the gradient norms, which is difficult in the composite case (but easy in the uncomposite case because this would be a simple consequence of smoothness and $\nabla f(x^\star)=0$). One can show that the real iterates and the virtual iterates are not far away from each other, i.e. $\|e_t\|^2$ goes to zero, but this does not translate to an upper bound on the distance between the objective values. For now, we are not aware of any method to fix these problems, and we believe that the solution may require rethinking the virtual iteration framework from the ground up or ditching this framework altogether. Our analysis in Appendix I is a step in the latter direction as the analysis there does not rely on virtual iterations at all. It would be interesting future work to try and improve the analysis in Appendix I.

---

> > ### Comment · Reviewer_hvZz · 2025-11-27
> >
> > Thank you for the detailed rebuttal and the revised manuscript. I re-read the paper and cross-checked the changes you mentioned in your response.
> >
> > **(1) Did you address my main concerns?**
> >
> > - **Prior composite-EF work and novelty claim.**
> >   You have substantially revised Appendix A and the introduction to incorporate and discuss EC-ProxSGD/EC-RDA (Qian et al., 2020), EC-LSVRG/Quartz/SDCA (Qian et al., 2021a), EF-BV (Condat et al., 2022), and Safe-EF/EF21-Prox extensions (Islamov et al., 2025; Fatkhullin et al., 2025). The introduction now explicitly states that existing composite-EF works either impose additional structural restrictions, cannot handle stochastic gradients, or have worse dependence on the compression parameter $\delta$. You also refined the core novelty statement to:
> >   $$\text{“we are the first to obtain rates for EF in the convex composite setting that matches the uncomposite counterpart,”}$$
> >   together with an explicit display of the rate
> >   $$\mathcal{O}\!\left(\frac{R_0^2\sigma^2}{n\varepsilon^2} + \frac{R_0^2\sqrt{\ell\sigma}}{\delta^2\varepsilon^{3/2}} + \frac{\ell R_0^2}{\delta\varepsilon}\right),$$
> >   and you clarify that “first strong convergence” refers precisely to matching the best-known uncomposite EF rates. This directly addresses my earlier concern that the “first” claim ignored prior composite-EF results.
> >
> > - **Related-work coverage.**
> >   The updated Appendix A now carefully positions your work relative to Qian et al. (EC-ProxSGD, EC-RDA, EC-LSVRG/Quartz/SDCA), Safe-EF, EF-BV, and EF21-based composite extensions, with a clear explanation of differences in setting (finite-sum vs your general stochastic setting), structural assumptions (bounded gradients, PL/KL conditions, constrained domains), and $\delta$-dependence. This is exactly the type of comparison I asked for and resolves the missing-related-work issue.
> >
> > - **Baselines and experimental justification.**
> >   In the rebuttal you explicitly argue that, under your setting, EC-ProxSGD with $p=0$ reduces to proximal EF up to an initial shift, and that EF-BV with $\lambda=\nu=1$ and $\nabla f(x^\star)=0$ reduces to proximal EF21. You also explain why EC-LSVRG/Quartz/SDCA target different finite-sum / primal-dual problems. In the revised paper, the experiments now compare EControl+DA against proximal EF, proximal EF21, and proximal EControl on the composite tasks. While this does not add all prior composite-EF baselines as separate curves, the reasoning that they collapse to the methods already plotted (in your particular stochastic composite setting) is now stated and technically plausible. I consider this concern largely addressed, though the experimental section would still benefit from a more extensive empirical study in future work.
> >
> > - **Virtual vs real iterates and clarity on the gap.**
> >   In the revised version, the discussion of real vs virtual iterates in the main text and Appendix I is more explicit. You emphasize that your main $O(1/(n\varepsilon^2))$-style stochastic term and $\delta$-dependence guarantees are for (a random sample of) virtual iterates, and you spell out that the real-iterate bound has a weaker $\delta$-dependence and no linear speedup. The conclusion and experiments now clearly state that narrowing this gap is an open problem, and Fig. 1 empirically documents that real and virtual iterates are very close in practice. In your rebuttal you also explain the technical barrier: in the composite case one cannot simply bound $\|\nabla f(x_t)\|^2$ via smoothness, so controlling the objective gap from $\|e_t\|$ is nontrivial. This does not remove the theoretical gap, but it addresses my request for a clearer and more honest discussion.
> >
> > - **Assumptions and scope of the main theorems.**
> >   Section 2 and Section 4 now lay out the convexity, smoothness, stochastic-noise assumptions and precisely state the dependence on $n$, $\delta$, and $\sigma^2$ in the final complexity, with the rate in the introduction consistent with the theorem statements. This resolves my concern about the abstract claiming more generality than the theorems.
> >
> > - **Inexact dual averaging vs EC-RDA analysis.**
> >   The updated related-work discussion makes clear how your inexact DA template differs from the EC-RDA analysis: EC-RDA is finite-sum, assumes bounded gradients of objectives and regularizers, and has cubic dependence on $1/\delta$, whereas your framework works in a general stochastic convex composite setting without those bounded-gradient assumptions and achieves $\delta$-dependence matching uncomposite EF21/EControl rates. Together with the more structured presentation in Appendix D and I, this adequately answers my question about the mathematical novelty of the DA analysis.

---

> > > ### Comment · Reviewer_hvZz · 2025-11-27
> > >
> > > I also checked that the specific changes you promised in the rebuttal (updated Appendix A with Qian/EF-BV/Safe-EF and EF21-Prox discussions, clarified novelty statement in the abstract, more detailed EControl analysis in Appendix G, and explicit highlighting of the real-vs-virtual gap in Appendix I) are indeed present in the revised PDF.
> > >
> > > **(2) What remains unaddressed or only partially addressed?**
> > >
> > > - The **theoretical guarantees for the real iterates** remain strictly weaker than for the virtual iterates, with an $O(1/(\delta^4\varepsilon^2))$-type dependence that neither enjoys linear speedup in $n$ nor the $\delta$-free leading term. You now clearly acknowledge this and position it as an open question, which I appreciate, but the limitation itself remains.
> > > - The **experimental section** is still relatively modest in scope and does not include separate implementations of EC-ProxSGD/EC-RDA/EF-BV or variance-reduced composite-EF methods. Your argument that, instantiated in your setting, these reduce to proximal EF / proximal EF21 is reasonable and now clearly recorded, so I no longer see this as a blocking issue, but the empirical coverage is not as comprehensive as one might wish for such a theoretically ambitious paper.
> > >
> > > Overall, I feel that you have **substantially addressed the core concerns** in my original review: the novelty claims are now properly scoped relative to prior composite-EF work, the related-work section is much more complete and technically accurate, the inexact DA contribution is better positioned, and the limitations of the real-iterate analysis are clearly explained rather than glossed over. The remaining issues are, in my view, secondary and more in the category of “directions for future strengthening” than fundamental flaws.
> > >
> > > Accordingly, I am **raising my overall score to 6**.

---

### Official Review · Reviewer_fp7G · 2025-10-31

**Soundness:** 3
**Presentation:** 3
**Contribution:** 3
**Rating:** 8
**Confidence:** 3

**Summary:**

This paper presents a novel convergence analysis for distributed algorithms that incorporate error feedback mechanisms, referred to as **EControl**, to address **composite optimization problems**. The analysis is based on the **dual averaging framework**, which enables bounding the difference between the virtual and true iterates generated by the algorithms. This bound is a crucial step in demonstrating the advantages of error feedback, such as improved solution accuracy in algorithms using communication compression.

The paper establishes the convergence guarantees for distributed proximal algorithms using **EControl** under standard assumptions, including **Lipschitz smoothness** of the objective function and **unbiased**, **variance-bounded** stochastic gradients. Moreover, the **iteration complexity** of these algorithms for composite (constrained) optimization problems matches that of their unconstrained counterparts.

Finally, empirical results validate the effectiveness of distributed proximal algorithms with **EControl**, showing that they achieve **linear speed-up** with respect to the number of participating agents.

**Strengths:**

This paper addresses an important gap in the study of **error-feedback algorithms** for distributed optimization by proposing novel analytical tools for handling **composite optimization problems** that involve the **proximal operator** under **communication constraints**. The authors effectively leverage the **dual averaging framework** to establish novel convergence analysis. In particular, Section 3 demonstrates how bounding the difference between the virtual and true iterates—a key aspect of analyzing error-feedback algorithms for unconstrained problems—can be extended to the composite setting.

The theoretical analysis is built upon standard and well-justified assumptions commonly used in gradient-based algorithms with communication compression, including:

1. **Convexity and Lipschitz smoothness** of the objective function
2. **$\alpha$-contractive** compression operators
3. **Unbiased** and **variance-bounded** stochastic gradients

Extensive literature on error feedback and communication compression, which is closely related to this paper, has been included in **Appendix A**.

The empirical evaluation is comprehensive. The proposed methods are tested on both synthetic data (softmax with $\ell_1$-regularization) and real data (logistic regression with $\ell_1$-regularization on FashionMNIST). Key findings include:
- The proposed algorithms support a significantly **larger step size** compared to baseline methods, which likely contributes to its superior convergence behavior.
- The proposed algorithms exhibit **linear speed-up** with respect to the number of clients, confirming their scalability and efficiency in distributed settings.

**Weaknesses:**

Algorithm 2 requires access to $\bar{g}$ in Line 16 to execute, which is **impractical**, even though full gradient communication occasionally occurs in Line 4. The key concern is that existing error-feedback methods for unconstrained problems—such as **EControl** and **EF21**—do **not** depend on periodic full gradient communication, making Algorithm 2 comparatively less communication-efficient.

Moreover, the empirical results in this paper simulate stochastic gradients by adding Gaussian noise to the true gradients. A more realistic setup would be to construct the problems such that stochastic gradients are obtained through **minibatching** over the local training samples available to each client, rather than by artificially injecting noise. This modification would allow for evaluating the impact of minibatched training samples on the convergence behavior of the proposed algorithms.

**Questions:**

1. I believe Theorem 4.4 indicates that the iteration complexity of **EControl with Dual Averaging (Algorithm 2)** for composite problems matches that of the **vanilla EControl** method for quasi-convex unconstrained problems, as presented in *Gao et al. (2024a)*. Could the authors please verify this point? If confirmed, emphasizing this result would further strengthen the paper’s contribution by highlighting the efficiency of the proposed convergence analysis techniques.

2. $m$ is not defined on Pages 2 and 4. Could the authors clarify what $m$ represents?

3. On Page 6, the definition of $A_t$ is unclear. I assume it is given by $A_t = \sum_{\tau \leq t} a_\tau$. Please confirm or clarify this notation.

4. In Section 5, the authors mention that “we simulate the stochastic gradient by adding Gaussian noise to the gradients.” What is the **standard deviation** of the Gaussian noise used in these experiments? This information is missing from the main text.

5. **Assumption 2.5 vs. Assumption 3.1:**  The weaker condition in Assumption 2.5,  i.e. $\mathbb{E}\|\| g_i(x;\xi^i) - g_i(y;\xi^i) \|\|^2 \leq \ell^2 \|\| x - y \|\|^2$,  seems potentially redundant with Assumption 3.1. Is it possible to remove Assumption 3.1 from the analysis, or are both required for specific parts of the proof?

---

> ### Author Response · Authors · 2025-11-25
>
> We thank the reviewer for the positive feedback on our paper and we address the reviewer's concerns in the following:
>
> - First, we would like to clarify that Algorithm 2 does **NOT** require periodic communication of full vectors. In particular, it only needs the clients to send $\bar g^i$ to the server only once at the end of the algorithm. The probabilistic approach at Line 4 of Algorithm 2 is part of Algorithm 3, which is a procedure to perform sampling on a stream of data consistently for all clients. It only requires the server to send the random bit to the clients in each iteration, and the client to perform only one full communication at the end of the procedure, and nothing else. As we have detailed in Theorem 4.4, Algorithm 2 only requires at most 3 rounds of full communication throughout the algorithm (regardless of the target error $\varepsilon$), which is asymptotically the same as EControl and EF21, both of which require 1 round of full communication at the start as well. The 2 rounds of additional full communication performed by our method come from one initial gradient step, and the communication of $\bar g^i$ at the end. These are mostly theoretical devices and not critical in practical settings.
>
> - We also wish to clarify that in our FashionMNIST experiment in Appendix B, we do not use simulated noise for the gradient and use the usual minibatch setup. In the synthetic experiment presented in the main text, we utilize artificial noise to precisely control the noise level and more accurately demonstrate the algorithm's behavior.
>
> - Indeed, the complexity of EControl with Dual Averaging matches that of vanilla EControl in the uncomposite setting asymptotically. We have clarified this in the updated version of the paper.
>
> - We thank the reviewer for catching this issue with the notation of $m$. We make the assumption that one full communication is $m$ times more expensive than one compressed communication in Page 4. In Page 2, we used $m$ by mistake to denote the number of constraints for the objective function that Safe-EF paper considered. We will update the notation on Page 2 and consistently use the meaning of $m$ as defined on Page 4 throughout the paper.
>
> - We thank the reviewer for catching this. We defined $A_t$ in Algorithm 3 but had to move it to the appendix due to space constraints. We have added the definition of $A_t$ back into the main paper as part of the update.
>
> - We use $\sigma^2 =25$ for both the linear speedup experiment and the virtual vs real iterates experiment. These are detailed in the corresponding paragraphs of each experiment. We will move it to the setup explanations for better visibility since both experiments used the same variance.
>
> - We thank the reviewer for the questions on Assumption 3.1 and would like to make the following clarification: in Section 3, we do not make use of Assumption 2.5 at all because in this section we do not even assume that we are using a distributed computing setting. To obtain our analysis of the general inexact dual averaging framework, we only need Assumption 2.3, 2.4 and 3.1. Assumption 3.1 is just a noise assumption for the stochastic gradients used by Algorithm 1. Assumption 3.1 can be instantiated in the distributed setting via Assumption 2.1. Assumption 2.5 is a smoothness assumption of the local functions which is only used in the distributed setting (Section 4).

---

### Official Review · Reviewer_bcYo · 2025-11-01

**Soundness:** 3
**Presentation:** 3
**Contribution:** 3
**Rating:** 8
**Confidence:** 3

**Summary:**

The paper studies Error Feedback (EF) methods for distributed composite optimization problems.
Existing EF analyses focus only on smooth (non-composite) settings, since the usual virtual-iterate technique used for EF fails when a non-smooth regularizer is present.
The authors address this issue by replacing standard gradient descent with a dual-averaging approach, which restores the additive structure needed for EF analysis.
By combining this idea with the EControl mechanism (a state-of-the-art EF variant), they obtain the first convergence guarantees for EF in the composite setting and achieve state-of-the-art iteration complexity.

**Strengths:**

1. The paper is well written and clearly organized.

2. The motivation is intuitive: the authors reformulate the EF method using dual averaging, which restores the additive structure (gradients plus accumulated compression error) and enables virtual-iterate analysis in the composite case.

3. The theoretical results are strong, with convergence rates matching the best-known results for the non-composite (smooth) setting.

**Weaknesses:**

1. It would be better if the paper provided more intuition about the **EControl** mechanism, since it plays a central role in the proposed method.
2. There are several typos and minor presentation issues:
   - Double $\coloneqq$ in Algorithm 1.
   - $A_T$ and $\tau_t$ are not defined.
   - $R_0$ is also not defined; it would be good to cite the work whose rate the paper matches in the non-composite case.
   - Several displayed equations are missing commas or periods at the end.

**Questions:**

Is it possible to extend EF21 to the composite setting instead of EF? Did you try?

---

> ### Author Response · Authors · 2025-11-25
>
> We thank the reviewer for the positive feedback on our paper and we address the reviewer's concerns in the following:
>
> - We have updated the paper to include more discussions on EControl in Appendix G. In short, EControl blends the classical error feedback mechanism and the gradient difference compression mechanism together, using a special control parameter $\eta$ for the error feedback strength.
>
> - We thank the reviewer for pointing out the typos and we have fixed them in the update.
>
> - It is indeed possible to extend EF21 to the composite setting as well, but probably not under our analysis framework because EF21 does not follow the classical error feedback template. Fatkhullin et al. (2021) analyzed a proximal variant of EF21, but it only applies to the non-convex and full gradient regime. Condat et al. analyzed EF21 in the composite setting under PL or KL conditions, again with full gradients; however, it's unclear whether their techniques can be extended to the general convex setting, which we consider. Islamov et al. (2025) provided a convex analysis of EF21 in their Safe-EF paper, but pointed out that their analysis critically relies on the compactness of the constraint set. Please refer to the updated related work section in Appendix A for a more detailed discussion.

---

### Meta-Review · Area_Chair_uTN2 · 2025-12-26

**Summary:**

This work presents a theoretical study of distributed stochastic optimization problems with composite structures under communication compression and error feedback mechanisms. The authors developed an analysis technique leading to iteration complexity that asymptotically matches the uncomposite counterpart. This is achieved by a novel method that combines Dual Averaging with the EControl mechanism. The authors complemented their theoretical result with experiments with logistic regression problems with L1 regularization.

Reviewers raised several concerns and clarifying questions regarding the presentation and the results of the paper, including providing more intuition about the core EControl mechanism and possible extension to EF21, whether full gradient communication is happening periodically in Algorithm 2, overstated novelty and correctness for some statements regarding the prior work and baselines, and limited experimental validation.

**Reviewer Concerns:**

Most of the concerns were addressed by the authors during the rebuttal/discussion phase. One concern that still remains is the experimental validation of the approach beyond synthetic experiments. However, weighing the theoretical nature and substantial contribution of the paper against the limited experimental validation, I recommend acceptance.

**Reviewer Scores:**

The reviewers were able to participate in the discussion with the authors and adjusted their scores accordingly. Particularly, Reviewer nFER acknowledged keeping the positive score unchanged, while Reviewer hvZz raised the initial score.

---

### Decision · Program_Chairs · 2026-01-26

Accept (Poster)